# Solver-Informed RL: Grounding Large Language Models for Authentic Optimization Modeling

**Yitian Chen**[1]*, **Jingfan Xia**[2,1]*, **Siyu Shao**[1,3], **Dongdong Ge**[4]†, **Yinyu Ye**[4,5]

[1] Cardinal Operations, China
[2] Shanghai University of Finance and Economics
[3] The University of Hong Kong
[4] Antai School of Economics and Management, Shanghai Jiao Tong University
[5] Department of Management Science and Engineering, Stanford University
`chenyitian@shanshu.ai`, `jf.xia@163.sufe.edu.cn`,
`siyu_shao@connect.hku.hk`, `ddge@sjtu.edu.cn`, `yyye@stanford.edu`

## Abstract

Optimization modeling is fundamental to decision-making in fields such as supply chain management, logistics, and financial engineering, but its complexity presents a major barrier to adoption. Automating model creation from natural language is key to improving efficiency and access. However, while Large Language Models (LLMs) are a promising tool for this, they often produce flawed or infeasible results due to errors and hallucinations. To address this issue, we propose Solver-Informed Reinforcement Learning (SIRL), a framework that uses Reinforcement Learning with Verifiable Reward to improve LLMs ability to generate accurate and executable optimization models. Specifically, SIRL automatically assesses the executable code and the instance-level mathematical model represented by the associated `.lp` files. This process yields precise feedback on syntactic validity, feasibility, and solution quality, which serves as a direct reward signal to guide the reinforcement learning process. Furthermore, this verification mechanism also supports our instance-enhanced self-consistency method for creating high-quality training data. Extensive experiments on diverse public benchmarks demonstrate that models trained with our SIRL framework achieve state-of-the-art performance, substantially outperforming existing methods in generating accurate and executable optimization models. Specifically, our SIRL-32B model surpasses DeepSeek-V3 and OpenAI-o3 on the majority of these benchmarks. Our code is publicly available at https://github.com/Cardinal-Operations/SIRL.

## 1 Introduction

Optimization modeling provides a powerful framework for decision-making across diverse fields, from logistics and finance to engineering and machine learning [1, 2, 3, 4, 5]. The process involves two primary steps: first, converting a natural language problem description into a formal mathematical model, and second, employing a solver to solve an optimization problem to get the optimal solution. Despite the maturity and powerful capabilities of modern optimization solvers, such as Gurobi [6], COPT [7] and CPLEX [8], formulating these intricate real-world problems into precise mathematical models and executable codes remains a significant bottleneck, often requiring substantial domain expertise and considerable manual effort [2, 9].

---

*Both authors contribute equally to this research.
†Corresponding authors.

39th Conference on Neural Information Processing Systems (NeurIPS 2025).

To bridge this gap, modeling languages such as MiniZinc [10] and Essence [11] emerged, offering a high-level declarative syntax to abstract away low-level solver details, particularly for constrained programming. However, even with these languages, the crucial first step, transforming a natural language problem description into a correct and efficient model, remains a formidable manual task [12]. The advent of Large Language Models (LLMs), (e.g., GPTs [13], Gemini [14, 15], Deepseek [16]), offers a promising avenue to automate or assist in this mathematical modeling and code generation process, potentially democratizing access to optimization solvers. However, ensuring the correctness, feasibility, and solver-compatibility of LLM-generated optimization models presents a significant and increasingly active research challenge. In general, existing approaches that leverage LLMs for optimization modeling can fall broadly into two categories. Firstly, prompt-based or agent-based approaches utilize the frozen capabilities of powerful foundation LLMs [13, 15, 14, 16], guide the models to extract relevant problem information, and generate the corresponding mathematical models and executable codes for optimization solvers [17, 18, 19, 20]. Although accessible, these methods do not adapt the underlying model parameters and can be sensitive to prompt design and the chosen foundation LLMs.

A distinct line of work addresses training open-source LLMs [21, 22] to enhance their capabilities for optimization modeling using offline learning approaches. This encompasses techniques such as Supervised Fine-Tuning (SFT) on demonstrations data, and alignment techniques (e.g., DPO [23], KTO [24]) on preference data. The success of existing offline methods [25, 26, 27] hinges on the availability of carefully curated datasets. These datasets, comprising problem descriptions paired with either detailed mathematical models and code demonstrations or comparative human preference labels, are typically generated through human annotation [27] or synthesis [25, 26]. Training on these curated datasets enables offline learning approaches to capture the data's stylistic and structural patterns, such as mathematical formulation and solver code, and achieve good performance. However, since their training objective focuses on mimicking demonstrations or aligning preferences, these methods still struggle to inherently guarantee functional correctness or solution feasibility, which are essential for reliable solver execution.

Recently, the development of powerful Large Reasoning Models (LRMs) marks a major leap in the reasoning capabilities of LLMs [28, 29, 30]. The process begins with Chain-of-Thoughts (CoTs) prompting [31], which unlocked the ability for LLMs to generate explicit, multi-step reasoning paths. Building on this, Reinforcement Learning with Verifiable Rewards (RLVR) [29, 32] provides the essential feedback mechanism to ensure the correctness of these paths. RLVR directly optimize LLMs' policies using objective feedback from verified outcomes, such as the validation of mathematical solutions or the execution of generated code against test cases [32, 29, 30, 33, 34]. This powerful combination has led to LRMs [28, 29, 30] that excel at complex reasoning, demonstrated by high performance on challenging mathematical competitions like the AIME and the International Mathematical Olympiad (IMO) [29, 28].

For optimization tasks, solving a real-world problem involves a multistep process encompassing problem analysis and reasoning, mathematical modeling, and code implementation [17, 18, 19, 20, 25, 26, 27], resembling a CoT reasoning process. The outputs of this process–the mathematical model and the solver code–can be verifiable using external optimization solvers. Verification involves steps such as syntax checks, feasibility assessment through model solving, and comparison of the objective value against known optimal value. These verifiable checks yield objective and richness reward signals, enabling RLVR [35, 29, 32, 36] to directly optimize the LLM generation towards producing correct, feasible, and high-quality outputs for complex optimization problems.

To the best of our knowledge, this is the first application of RLVR to directly enhance LLMs' proficiency in optimization modeling. The rewards, including feasibility status, objective function value, and mathematical model statistics from the `.lp` file, are obtained by executing the generated code. Specifically, the `.lp` file, which is a standard format for optimization solvers, provides an explicit instance-level representation of the generated mathematical model, implicitly reflecting the LLM's reasoning process. This allows us to move beyond simple, results-based rewards and design a more refined, process-aware reward [37]. By combining common solver outputs with detailed statistics from the `.lp file`, we ingeniously blend the correctness of the outcome-based reward with the fine-grained nature of the process-based reward. This enables a solver-informed evaluation that ensures accurate assessment of the LLM's performance and validity on both mathematical modeling and outcome correctness.

Our main contributions are fourfold: (1) We introduce a simple yet effective instance-enhanced self-consistency method for synthesizing high-quality training data for optimization tasks. (2) We introduce SIRL, **solver-informed reinforcement learning**, an automated RLVR framework for LLMs in optimization modeling with a novel surrogate function. By enabling a balance between diverse reasoning exploration and the requirements for accuracy and validity in mathematical models and code, this function leads to a significant improvement in the authenticity and reliability of the generated optimization solutions. (3) We demonstrate how classical optimization solvers can serve as effective and powerful tools for both enhancing the data synthesis process and providing rich reward signals for the proposed SIRL framework. (4) Through extensive experiments on diverse public benchmarks, our SIRL-trained 7B model achieves state-of-the-art performance, significantly outperforming existing offline learning and agent-based methods in generating correct and reliable models. Moreover, our larger 32B model establishes new benchmarks by surpassing powerful baselines like DeepSeek-V3 and OpenAI-o3 on the majority of these benchmarks.

## 2 Related work

Our work builds upon and contributes to several research areas, primarily LLMs for optimization, synthetic data generation for LLM training, the paradigm of reinforcement learning with tool-verified feedback mechanisms.

**LLMs for optimization modeling.** The application of LLMs for optimization modeling has emerged as a prominent research direction. Early work mainly relied on prompt engineering techniques, including agent-based prompting [19] and multi-agent reasoning [18, 20], but was limited by careful prompt design and the capability of the foundation LLMs. More recent work focuses on offline learning approaches to adapt LLM parameters using specialized mathematical modeling datasets. For instance, ORLM [25] and OptMATH [26] employed Supervised Fine-Tuning (SFT) with datasets constructed via semi-automated synthetic data generation workflows, LLMOPT [27] introduced an alignment learning framework with multi-instruction tuning and a self-correction mechanism.

**LLM-Based data synthesis.** Fine-tuning LLMs on specialized tasks necessitates high-quality datasets, a resource-intensive requirement often necessitating domain expertise. Data synthesis offers a scalable solution. Examples of general-purpose synthesis techniques include self-instruction [38] and the WizardInstruct series [39, 40, 41]. Specifically, to improve reasoning capabilities, recent work focuses on synthesizing reasoning trajectory data, exemplified by the work Star [42], rstar [43], and Phi [44], aiming to enhance complex reasoning ability in tasks such as math and code. Within the domain of optimization modeling, several approaches have been explored: ORLM [25] introduced a semi-automated method for synthesizing operations research datasets based on self-instruct frameworks [38]; ReSocratic [45] employs reverse data synthesis via back-translation; and OptMATH [26] generates controllable complexity data with verification.

Ensuring correctness and formal adherence is a central challenge when synthesizing high-quality data for tasks requiring rigorous output. Techniques addressing this include LLM-as-a-judge [46], which leverages foundation models for evaluation, rejection sampling [47, 26], which provides automated verification, and self-consistency [38], which relies on multi-sample voting for robustness.

**Reinforcement learning with verifiable reward.** Reinforcement Learning from Human Feedback (RLHF) [48] marked a significant step in aligning LLMs, typically using reward models trained on human preferences [49, 50]. However, for tasks where the desired output has objectively verifiable properties, such as mathematical reasoning [51, 50, 52] and code generation [53, 54], relying solely on subjective human preference is suboptimal and can struggle with objective correctness and potential reward hacking [55, 56, 57]. For example, a model might generate code with correct syntax but a flawed function. RLVR has demonstrated significant success in enhancing LLMs' performance across these domains, as evidenced by strong results on benchmarks like GSM8K [50], MATH [53], AIMO [52], HumanEval [53], CodeForce [54], and has been a key technique in developing highly capable large reasoning models (e.g., OpenAI-O1 [28],DeepSeek-R1 [29], Kimi-k1.5 [30]) and state-of-the-art coding models (e.g., Tulu-3 [32]).

**External tools as verifier.** Leveraging external tools or formal verification mechanisms to validate LLM-generated structured outputs, particularly in domains that require high fidelity and correctness, is becoming an increasingly critical area of research [58, 36, 59]. In mathematical theorem proving,

the Lean proof assistant [60] acts as a verifier for LLM-generated proofs [59, 58, 61, 62]. Googles AlphaGeometry series [63, 64] combines reasoning LLMs with symbolic engines DDAR, achieving breakthroughs in the IMO-level mathematical tasks. For formal domains like mathematics and programming that demand high output fidelity, integrating code interpreters is commonly employed to improve LLMs' performance and ensure correctness. The basic approach involves generating complete solutions and validating them post-hoc via external execution using interpreters or compilers [53, 65]. Feedback from these validation checks provides a signal for model refinement. More advanced methods either integrate tool interactions directly within the LLMs' reasoning process (e.g., ToRA [66], MARIO [67]) or focus on enabling the LLMs to learn effective strategies for using tools autonomously (e.g., START [33], ReTool [34]).

Our work proposes a direct analogy: just as Lean verifies mathematical proofs and code compilers verify code for general math and coding problems, classical optimization solvers [6, 7, 8, 68] serve as the natural, powerful, and domain-specific objective oracles, which can both enhance the synthesis of high-quality optimization task data and provide rich reward signals for the proposed RLVR framework.

## 3   Method

Let the training dataset be $\mathcal{D} = \{(x_i, y_i^*)\}_{i=1}^N$, where $x_i$ represents the natural language description of the i-th optimization problem and $y_i^*$ is the corresponding ground-truth optimal objective function value. We model the problem solver as an LLM policy $\pi_\theta$, parameterized by $\theta$. Given an input problem description $x$, the policy $\pi_\theta$ generates a response $\mathbf{z}$ containing sequences of reasoning process leading to an objective function value $y$, derived via a mapping function $g(x, \mathbf{z})$.

To guide the learning process, we introduce the SIRL framework, which incorporates a verifiable reward function $r(x, \mathbf{z}, y^*)$. This function quantifies the quality of the derived objective value $y$ for the problem $x$, using the ground truth $y^*$ as a reference. Our goal is to optimize the policy parameter $\theta$ to maximize the expected reward:

$$\max_\theta \mathbb{E}_{(x,y^*)\sim\mathcal{D}, \mathbf{z}\sim\pi_\theta(\cdot|x), y\sim g(x,\mathbf{z})}[r(x, \mathbf{z}, y^*)]. \tag{1}$$

In the following subsection, we outline the key components of our framework: the data synthesis pipeline used to construct the training dataset, the solver-informed reinforcement learning method with its surrogate function design tailored for optimization tasks and the two-stage training curriculum including reward design.

### 3.1   Data synthesis framework

**Overall framework.**  The data synthesis process starts from curated seed data and integrates LLM generation with subsequent steps including self-instruction, augmentation, evaluation, consistency assessment, and filtering. While our work primarily follows the OR-Instruct pipeline in ORLM [25], this approach contrasts with previous work synthesizing complete (question, model, code) sequences [25, 26, 69] by focusing on generating high-quality (question, answer) pairs, where the answer is the optimal value by executing an optimization solver.

As illustrated in Figure 1, we firstly sample questions from the curated seed data, combine them contextually with scenarios sampled from a predefined list, and generate new problems that remain structurally similar to the original. Subsequently, we implement an augmentation phase to increase the challenge related to semantic interpretation, mathematical modeling complexity, and solution complexity, obtaining a larger corpus of extended questions. Then, the LLM-as-a-judge approach [46] validates the generated problems for practical relevance and semantic consistency. Following the validation of the problem, the LLM is employed to generate mathematical models and corresponding executable code. This code is then executed to produce the objective value, feasibility status and the instance-level mathematical models represented by `.lp` files.

To ensure high correctness of the final answer, we assign multiple LLM roles (10 roles) per problem [70, 71] and apply a novel instance-enhanced self-consistency framework when generating answers. Furthermore, an iterative reflection and refinement process [72] is employed to address execution issues, regenerating or refining code upon errors, and regenerating the model and code when infeasible solutions are encountered.

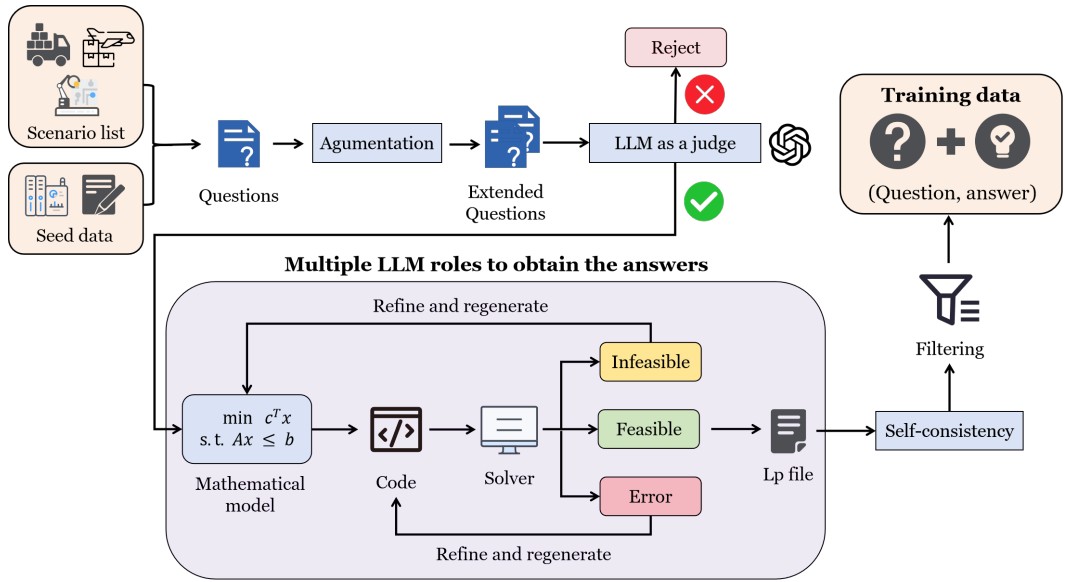

Figure 1: An overview of the data synthesis process.

Finally, guided by the principle "Less is More" [73, 74], we filter the (question, answer) pairs. Samples are excluded if a baseline Qwen-32B-Instruct model [22] achieves an 80% pass rate (8/10 attempts across different roles) in generating executable code matching the optimal value, as these instances are considered too elementary. The retained pairs are incorporated into the final training dataset.

In the next part, we detail our novel, simple yet effective instance-enhanced self-consistency method used within our data synthesis framework.

**Instance-enhanced self-consistency.** Relying solely on majority voting of final results for self-consistency in optimization modeling can be limiting, potentially ignoring embedded model information. We enhance this by integrating structural data extracted from the instance's `.lp` file. The `.lp` files are chosen as they formally encode key model properties (e.g., variable types, objective direction), providing a formalized, implementation-agnostic representation of the instance-level mathematical model. Specifically, after executing the generated code associated with a role $r$ and obtaining the corresponding `.lp` file, we extract the following features:

- $O_r$: The final objective function value.
- $D_r \in \{\max, \min\}$: The optimization direction (maximization or minimization).
- $N_{bin,r}$: The count of binary (0-1) variables.
- $N_{int,r}$: The count of general integer variables (distinct from binary variables).

These statistics provide detailed, model-level insights that supplement the final numerical outcome $O_r$. Let $R$ be the set of roles that generated responses for a given question. We assign a score $S(r)$ to the response of each role $r \in R$ using a weighted voting mechanism that measures consensus with other roles. We define a consensus function $\psi(X_r)$ for a feature $X \in \{O, D, N_{bin}, N_{int}\}$ as the count of the roles $r' \in R$ whose corresponding feature value $X_{r'}$ is identical to $X_r$:

$$\psi(X_r) = |\{r' \in R \mid X_{r'} = X_r\}|. \tag{2}$$

The final score $S(r)$ for the response from the role $r$ is calculated as a weighted sum reflecting consensus across the extracted features:

$$S(r) = w_1 \cdot \sqrt{\psi(O_r)} + w_2 \cdot \sqrt{\psi(D_r)} + w_3 \cdot \sqrt{\psi(N_{bin,r})} + w_4 \cdot \sqrt{\psi(N_{int,r})}. \tag{3}$$

The weights $(w_1, w_2, w_3, w_4)$ determine the relative contribution of the individual consensus components to the final result. In our current implementation, all weights are set to 1, giving equal importance to each consensus component.

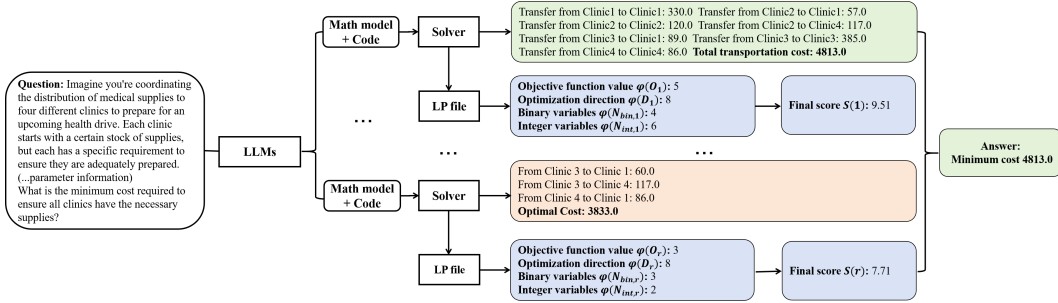

Figure 2: An overview of the data synthesis process.

Figure 2 provides an illustrative example of this enhanced self-consistency method. For a given question, multiple LLM roles generate "Math model + Code" trajectories. Each code is executed by a solver, producing a solution and the corresponding `.lp` file. From the `.lp` file, we extract structural features (objective value consensus $\psi(O_r)$, optimization direction $\psi(D_r)$, variable counts $\psi(N_{bin,r}), \psi(N_{int,r})$). The final score is calculated according to Equation 3. Finally, the objective value $O_{r^*}$ from the response achieving the highest score, where $r^* = \arg\max_{r \in R} S(r)$, is selected as the definitive answer to the question.

## 3.2 SIRL: Solver-Informed Reinforcement Learning

**RLVR for LLMs in optimization modeling.** In optimization tasks, to obtain the optimal value from a problem description $x$, a complex reasoning process is involved with distinct stages: first, analyzing the problem description $x$ to identify the key information, such as optimization problem type and its core components; second, constructing the mathematical formulation which typically involves the parameter set, objective functions, and constraints; and finally, generating the corresponding executable code, which is then executed by the optimization solver to produce the objective value $y$ and other relevant output (e.g., decision variable values and solution status). To address this complex reasoning process that integrates both mathematical modeling and code implementation, we utilize the Chain of Thought (CoT) [31] method, where the LLM policy $\pi_\theta$ generates a sequence of intermediate thoughts $\mathbf{z} = (\mathbf{z}^1, \mathbf{z}^2, \dots, \mathbf{z}^m)$ that serve as a crucial bridge between the initial problem $x$ and the final result $y$.

Specifically, a well-designed system prompt structures the sequence of thoughts $\mathbf{z}$ into segments reflecting the defined reasoning, modeling, and code generation stages: $(\mathbf{z}^1, \mathbf{z}^2, \dots, \mathbf{z}^{m-2})$ contains the analysis and reasoning process (e.g., the identification of optimization problem type, algorithmic choices, or reasoning steps towards the final model structure); $\mathbf{z}^{m-1}$ contains the concise mathematical modeling; and $\mathbf{z}^m$ contains the derived executable code. The final value $y$ is obtained deterministically by extracting and executing code in $\mathbf{z}^m$, formally represented as $y = g(x, \mathbf{z})$, where $g$ denotes the deterministic code execution function.

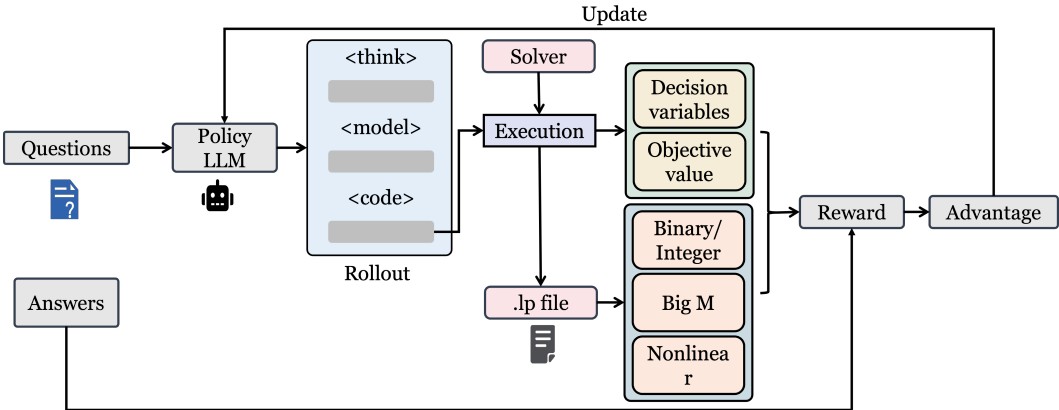

Figure 3: An overview of the SIRL training framework.

At the token level, each thought $\mathbf{z}^j$ is realized as a sequence of tokens $\mathbf{z}^j = (z_1^j, \ldots, z_{T_j}^j)$. The token $z_t^j$ within this sequence is sampled autoregressively from the model's policy $\pi_\theta(\cdot|x, \mathbf{z}^1, \ldots \mathbf{z}^{j-1}, z_1^j, \ldots, z_{t-1}^j)$, conditioned on the initial input $x$, all previously completed thoughts $(\mathbf{z}^1 \ldots \mathbf{z}^{j-1})$, and all tokens generated so far in the current thought.

**Surrogate function design: Partial KL.** To maximize the expected verifiable reward objective defined in Equation (1), we employ REINFORCE++ [75], a robust policy gradient algorithm that incorporates key techniques from Proximal Policy Optimization [76].

In each training iteration, a batch of data $\{x_i, y_i^*\}_{i=1}^n$ is sampled from the training dataset. Then, for each $x_i$, the policy $\pi_\theta$ is used to sample a set of $K$ complete response trajectories $\{\mathbf{z}_{i,k}\}_{k=1}^K$, where each $\mathbf{z}_{i,k}$ is a sequence of tokens generated autoregressively and composed of a sequence of thoughts. For simplicity, we denote the collected batch as $\mathcal{B}$, where each tuple $(x, \mathbf{z}, y^*) \in \mathcal{B}$ consists of $x$, the input problem description, $\mathbf{z}$, the generated response trajectory, and $y^*$, the ground-truth objective value.

The algorithm updates the policy parameters $\theta$ by maximizing the following clipped surrogate objective:

$$\mathcal{J}^{\text{Reinforce++}}(\theta) = \frac{1}{|\mathcal{B}|} \sum_{(x, \mathbf{z}, y^*) \in \mathcal{B}} \left[ \frac{1}{\sum_{j=1}^m T_j} \sum_{j=1}^m \sum_{t=1}^{T_j} \min \left( \rho_t^j A_t^j, \text{clip} \left( \rho_t^j, 1 - \epsilon, 1 + \epsilon \right) A_t^j \right) \right],$$

where $\epsilon$ is the clipping hyperparameter; $\rho_t^j = \frac{\pi_\theta(z_t|x, \mathbf{z}^{<j}, \mathbf{z}_{<t}^j)}{\pi_{\theta_{\text{old}}}(z_t|x, \mathbf{z}^{<j}, \mathbf{z}_{<t}^j)}$ is the probability ratio of generating token $z_t$ under the new policy versus the reference policy $\pi_{\theta_{old}}$; and $A_t^j$ denotes the token-level advantage computed for token $z_t^j$.

Building on this algorithmic structure, the per-timestep reward signal $\hat{r}_t^j$ and its corresponding advantage $A_t^j$ that is normalized across the mini-batch are defined as follows:

$$\begin{aligned} \hat{r}_t^j &= \mathbb{I}(z_t^j = [\text{EOS}])r(x, \mathbf{z}, y^*) - \beta \text{KL}(j, t), \\ A_t^j &= (\hat{r}_t^j - \mu_{\hat{r}_t^j})/\sigma_{\hat{r}_t^j}, \end{aligned} \tag{4}$$

where $\mathbb{I}(s_t^j = [\text{EOS}])$ is an indicator function that assigns the reward $r(x, \mathbf{z}, y^*)$ only when $z_t^j$ is the end-of-sequence token. A token-level KL penalty component, $\text{KL}(j, t)$, is included to penalize policy deviations from the reference policy $\pi_{old}$.

To reconcile the tension between exploratory reasoning trajectories diversity (which may deviate significantly from the reference model distribution) and strict adherence to mathematical formulation/solver syntax requirements in optimization tasks, we propose `Partial KL`. This novel design selectively applies the KL penalty to the mathematical formulation $\mathbf{z}^{m-1}$ and solver code $\mathbf{z}^m$ segments. The value for the KL term, $\text{KL}(j, t)$, within these segments is computed using the unbiased estimator described in [77]:

$$\text{KL}(j, t) = \begin{cases} \rho_t^j - \log \rho_t^j - 1 & j \in \{m-1, m\}, \\ 0 & \text{otherwise.} \end{cases} \tag{5}$$

The detailed training process is illustrated in Figure 3.

The rationale for the `Partial KL` design that utilizes selective KL regularization is twofold:

1. **Exploration in reasoning**: For reasoning steps $(\mathbf{z}^1, \mathbf{z}^2, \ldots, \mathbf{z}^{m-2})$, the KL penalty is omitted. This fosters exploration, enabling the policy to better understand the problem background and identify diverse reasoning paths and implicit constraints [78, 79].

2. **Stability in modeling and code generation**: For the critical mathematical formulation $\mathbf{z}^{m-1}$ and solver code $\mathbf{z}^m$ segments, the KL penalty ensures the generated output remains well-structured and adheres to expected formats, preventing policy collapse while still allowing gradual improvement guided by the reward.

Our SIRL framework, which incorporates `Partial KL`, enables the policy to achieve effective exploration and understanding of the problem while improving code execution accuracy, and yields high-quality outputs for optimization tasks.

## 3.3 Reward design and training scheme

The success of our SIRL framework hinges on its verifiable reward function, implemented as a staged, rule-based system [29, 32]. Integrated with the optimization solver, this system provides objective verification signals used within a two-stage curriculum [80, 81] to progressively train the model and enhance its optimization modeling capabilities.

Given a question $x$, generated trajectories $\mathbf{z}$, ground-truth answer $y^*$, the two-stage reward function $r(x, \mathbf{z}, y^*)$ is defined as follows:

$$r(x, \mathbf{z}, y^*) = \begin{cases} R_{\text{format}}(\mathbf{z}) + R_{\text{exec}}(\mathbf{z}) + R_{\text{accur}}(x, \mathbf{z}, y^*) & \text{Stage-1,} \\ R_{\text{format}}(\mathbf{z}) + R_{\text{exec}}(\mathbf{z}) + R_{\text{accur}}(x, \mathbf{z}, y^*) + R_{\text{bonus}}(x, \mathbf{z}, y^*) & \text{Stage-2.} \end{cases} \quad (6)$$

In stage-1, we focus on building the model's fundamental capabilities in formulating and solving standard optimization problems. The reward function comprises three key components: format, execution, and accuracy. Emphasis is placed on the execution component via additional incentives for correctly executed code. This ensures that the generated models are both mathematically sound and executable.

Building on fundamental capabilities, Stage 2 aims to tackle more complex problems through the bonus reward $R_{\text{bonus}}$, which is based on the generated mathematical model associated with the LP file and designed to incentivize advanced modeling techniques (e.g., Big-M, nonlinear formulations) which are crucial for complex and challenging problems. This bonus is granted only when two conditions are met: (1) the generated solution is correct, and (2) it incorporates advanced modeling strategies. The complete reward function formulation is detailed in the Appendix.

# 4 Experiments

## 4.1 Main results

Table 1: Performance comparison of models on benchmarks.

| Types | Models | Acc (pass@1) | | | | | Macro AVG |
|---|---|---|---|---|---|---|---|
| | | NL4OPT | MAMO Easy | MAMO Complex | IndustryOR | OptMATH | |
| Baseline | GPT-4 | 89.0%* | 87.3%* | 49.3%* | 33.0%* | 16.6%* | 55.0%* |
| | DeepSeek-V3.1 | 84.8% | 88.9% | 63.5% | 44.0% | 43.9% | 65.0% |
| LRMs | DeepSeek-R1 | 82.4% | 87.2% | **67.9%** | **45.0%** | 40.4% | 64.6% |
| | OpenAI-o3 | 69.4% | 77.1% | 51.2% | 44.0% | 44.0% | 57.1% |
| Agent-based | OptiMUS | 78.8%* | 77.2%* | 43.6%* | 31.0%* | 20.2%* | 49.4%* |
| Offline-learning | ORLM-LLaMA-3-8B | 85.7%* | 82.3%* | 37.4%* | 24.0%* | 2.6%* | 46.4% |
| | LLMOpt-Qwen2.5-14B | 80.3%* | 89.5%* | 44.1%* | 29.0%* | 12.5%* | 51.1% |
| | OptMATH-Qwen2.5-7B | 94.7%* | 86.5%* | 51.2%* | 20.0%* | 24.4%* | 55.4% |
| | OptMATH-Qwen2.5-32B | 95.9%* | 89.9%* | 54.1%* | 31.0%* | 34.7%* | 61.1% |
| Online-RL | SIRL-Qwen2.5-7B | 96.3% | 91.7% | 51.7% | 33.0% | 30.5% | 60.6% |
| | SIRL-Qwen2.5-32B | **98.0%** | **94.6%** | 61.1% | 42.0% | **45.8%** | **68.3%** |

Values marked with * are from original or reproduced papers with the criterion: relative error $< 10^{-6}$.

To evaluate the effectiveness of the proposed SIRL framework, we developed two models at different scales, SIRL-Qwen2.5-7B and SIRL-Qwen2.5-32B. Both were initialized from their respective base instruction models, Qwen2.5-7B-Instruct and Qwen2.5-32B-Instruct [22], without any prior supervised fine-tuning. Performance was evaluated on four benchmarks- NL4OPT [82], MAMO [83], IndustryOR [25] and OptMATH [26] –using pass@1 accuracy. Consistent with the evaluation protocol proposed by OptMATH [26], a solution is considered valid if the relative error is less than 1e-6. A detailed description of these datasets and evaluation criterion is provided in Appendix B. Table 1 presents the main results.

**Performance of SIRL-Qwen2.5-7B.** Our SIRL-Qwen2.5-7B model consistently outperforms all existing 7B and 14B models trained with other offline learning methods [25, 27, 26], as well as the agent-based approach OptiMUS [19].

**Performance of SIRL-Qwen2.5-32B.** The SIRL-Qwen2.5-32B model demonstrated superior performance across all evaluated benchmarks. Despite its significantly smaller size, our 32B-parameter model achieved a higher Macro Average than much larger models, including the 671B-parameter foundation model Deepseek-V3.1 and powerful reasoning models like DeepSeek-R1 and OpenAI-o3.

These finds highlight the efficiency of our proposed SIRL mechanism in enhancing LLMs' abilities for optimization formulation and code solving, demonstrating its ability of tackling complex optimization modeling challenges. Details regarding our experimental setup, which include prompt templates, training hyperparameters, and decoding strategies, are provided in the Appendix for a complete overview.

## 4.2 Instance-enhanced self-consistency

Table 2: Performance of value-based and instance-enhanced self-consistency on Qwen2.5 Models.

| Metric | NL4OPT | | MAMOEasy | | MAMOComplex | | IndustryOR | | OptMATH | | Average | |
| | 7B | 32B | 7B | 32B | 7B | 32B | 7B | 32B | 7B | 32B | 7B | 32B |
|---|---|---|---|---|---|---|---|---|---|---|---|---|
| pass@1 | 65.7% | 67.8% | 81.9% | 83.7% | 17.1% | 26.5% | 19.0% | 23.0% | 4.1% | 16.6% | 37.6% | 43.5% |
| val_sc@5 | 69.8% | 70.2% | 85.3% | 86.0% | 25.1% | 35.1% | 25.0% | 32.0% | 5.7% | 22.3% | 42.2% | 49.1% |
| inst_sc@5 | 70.6% | 71.0% | 85.4% | 86.3% | 29.9% | 38.4% | 26.0% | 33.0% | 13.0% | 27.5% | 45.0% | 51.2% |
| **Diff (inst-val)@5** | **1.1%** | **1.1%** | **0.1%** | **0.3%** | **19.1%** | **9.4%** | **4.0%** | **3.1%** | **128.1%** | **23.3%** | **30.5%** | **7.5%** |
| val_sc@10 | 68.6% | 73.5% | 85.6% | 85.9% | 29.9% | 38.4% | 28.0% | 34.0% | 9.8% | 27.5% | 44.4% | 51.9% |
| inst_sc@10 | 69.0% | 72.2% | 85.7% | 86.0% | 32.2% | 39.3% | 30.0% | 36.0% | 16.6% | 34.2% | 46.3% | 53.5% |
| **Diff (inst-val)@10** | **0.6%** | **-1.8%** | **0.1%** | **0.1%** | **7.7%** | **2.3%** | **7.1%** | **5.9%** | **69.4%** | **24.4%** | **17.0%** | **6.2%** |

We evaluated different self-consistency approaches on the Qwen2.5 models [22] (7B-Instruct and 32B-Instruct) to assess the effect of leveraging instance-level information. The value-based self-consistency method (val_sc) is a direct adaptation of the standard self-consistency approach where the final score of different roles depends only on the final objective function value. The instance-enhanced self-consistency method (inst_sc) also includes structural information within the generated optimization models, augmenting the consensus mechanism. The consensus function of the final objective function value, optimization direction, the count of binary variables, and general integer variables are given the same weight in Equation 3.

Table 2 indicates that self-consistency through majority voting outperforms the baseline single-pass generation (pass@1). Both val_sc and inst_sc methods demonstrate consistently higher accuracy than the pass@1 baseline. Furthermore, a comparative analysis between the two self-consistency variants suggests that incorporating instance-level information (optimization direction, variable counts) into the voting mechanism provides a more robust measure of consensus, leading to improved selection of correct solutions compared to relying solely on the final objective value.

## 4.3 Ablation Study

In this section, we present a series of ablation studies to examine the impact of the surrogate function design based on the `Partial KL` strategy and the proposed two-stage reward mechanism.

### 4.3.1 Ablation study on different surrogate function designs.

Table 3: Ablation study on different surrogate function designs.

| Type | MAMO Complex | | IndustryOR | | OptMATH | |
| | Acc(pass@1) | ER | Acc(pass@1) | ER | Acc(pass@1) | ER |
|---|---|---|---|---|---|---|
| Partial KL | 51.7% | 98.1% | 33.0% | 96.0% | 30.5% | 92.2% |
| Full KL | 48.3%(↓3.4%) | 98.5%(↑0.4%) | 30%(↓3.0%) | 95.0%(↓1.0%) | 28.3%(↓2.2%) | 93.4%(↑1.2%) |
| Without KL | 47.3%(↓4.4%) | 95.6%(↓2.5%) | 29%(↓4.0%) | 87.0%(↓9.0%) | 29.5%(↓1.0%) | 80.1%(↓12.1%) |

We evaluated three distinct surrogate function designs: (i) `Full KL`: the standard approach applying full KL-divergence regularization against the reference policy; (ii) `Without KL`: an approach

omitting KL-divergence regularization, which is popular in RLVR training for mathematical problems [78, 79] such as AIME [52]; (iii) `Partial KL`: our novel design that applies the KL penalty selectively to the mathematical formulation and code segments.

Table 3 reports both the pass@1 accuracy and execution rate (ER), which measures the percentage of generated solutions that successfully compile and return a valid result, across three more challenging datasets. The results show that the proposed `Partial KL` approach achieves the best performance across all benchmarks. In contrast, the Without KL design exhibits a dramatically lower execution rate than the other two strategies. This lower rate stems from removing KL divergence: while promoting more diverse exploration, it can lead to introducing irrelevant constraints from the problem background, increasing invalid generations. `Partial KL` resolves this issue by applying KL selectively, improving the execution rate while preserving reasoning diversity. The full comparison on all benchmarks and a detailed qualitative analysis, including a case study, are presented in the Appendix.

### 4.3.2 Ablation study on reward design.

Table 4: Performance results of the ablation study on reward design.

| Reward Type | Acc (pass@1) | | | | |
| --- | --- | --- | --- | --- | --- |
| | NL4OPT | MAMO Easy | MAMO Complex | IndustryOR | OptMATH |
| Two-stage rewards | 96.3% | 91.7% | 51.7% | 33.0% | 30.5% |
| Stage-1 reward only | 96.7% (↑0.4%) | 88.8% (↓2.9%) | 46.8% (↓4.9%) | 27.0% (↓6.0%) | 28.9% (↓1.6%) |
| Stage-2 reward only | 92.2% (↓4.1%) | 89.6% (↓2.1%) | 49.3% (↓2.4%) | 28.0% (↓5.0%) | 33.1% (↑2.6%) |

**Ablation study on reward design.** We compared the performance of the proposed two-stage reward mechanism (6) against models trained using only the stage-1 reward and using only the stage-2 reward. As shown in Table 4, using only the stage-1 reward yielded comparatively strong results on simple tasks such as NL4OPT. This indicates that this reward enables the model to learn stable foundational skills for optimization tasks. Meanwhile, employing only the stage-2 reward, which includes a bonus component incentivizing the model to learn advanced strategies, achieves the best performance on the most challenging OptMATH dataset. However, this led to diminished performance on simpler tasks such as NL4OPT. Overall, the integrated two-stage reward mechanism successfully balanced the objectives of the individual stages, resolving the trade-offs observed with single-stage rewards, thereby achieving superior performance in most benchmark tasks, with the exception of the OptMATH dataset.

## 5 Conclusion

In this paper, we present SIRL, a novel RLVR framework addressing the challenge of generating authentic mathematical optimization models with LLMs. The core contributions of this work are its unique surrogate function design, `Partial KL`, which selectively applies KL divergence to mathematical formulation and code segments, and its two-stage reward system, which leverages optimization solvers for automated verification. The comprehensive signals derived from this verification were valuable for both RL training and enhancing our data synthesis. Extensive experiments showed that the SIRL-trained model achieved superior performance in generating accurate, well-formed optimization models compared to existing methods. More broadly, the proposed techniques are applicable to tasks requiring LLMs to balance exploring diverse reasoning with ensuring solution validity, particularly in tool-augmented tasks. However, challenges such as reward hacking persist, and performance on difficult benchmarks like IndustryOR and OptMATH remains limited even for advanced models. Our future work will therefore target these gaps through systematic error analysis and targeted improvements.

## Acknowledgments and Disclosure of Funding

This research is partially supported by the National Natural Science Foundation of China (NSFC) [Grant NSFC-72225009, 72394360, 72394365].

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

# A    Technical Background for Automated Optimization Modeling

The automated conversion of a natural language problem into an optimization model follows a multi-step pipeline, as illustrated in our work. This process transforms an unstructured text description into verifiable mathematical artifacts and then into a solved optimization problem. To fully appreciate the mechanisms behind our data synthesis and RL framework, we overview the core components below.

## A.1    Core Components

The pipeline relies on the interplay of several key components:

**Natural Language Problem**    A natural language problem is an unstructured textual description of a decision-making scenario, often presented as a word problem or real-world query (e.g., How should a factory allocate resources to maximize profit given limited labor and materials?). These problems are intuitive for humans but ambiguous and non-executable for computers. It's necessary to translate into formal mathematical representations.

**Mathematical Model**    A mathematical model formalizes a real-world decision problem into a structured optimization problem. It consists of three primary elements: (1) decision variables, which represent the unknowns to be determined (e.g., quantities to produce); (2) an objective function, a mathematical expression to maximize or minimize (e.g., profit or cost); and (3) constraints, which enforce limitations or requirements (e.g., resource availability or logical conditions).

**Optimization Solvers**    Solvers are the computational engines at the heart of optimization. They are highly specialized algorithms designed to find optimal solutions to formally defined mathematical problems. Their input is a precise mathematical model (variables, constraints, objective), and their output is a solution (if one exists).

**Modeling Libraries**    Modeling libraries are Application Programming Interfaces that provide a bridge between a general-purpose programming language (like Python) and the low-level solver. They allow developers to define variables, constraints, and objectives using familiar programming constructs, abstracting away the complexities of direct solver interaction. The code generated by the LLM in our work utilizes these libraries.

**Optimal Solution and Optimal Value**    The optimal solution refers to the set of values for the decision variables that satisfy all constraints and achieve the best possible value of the objective function (e.g., maximum profit or minimum cost). The optimal value is the corresponding objective function value at this solution point. For infeasible or unbounded problems, no optimal solution exists, and the solver indicates the status accordingly.

**The `.lp` File Format**    The `.lp` file is a standardized, human-readable text format for representing linear and mixed-integer programming problems. It serves as a universal blueprint of the mathematical model, independent of the programming language or library used to create it. This solver-agnostic nature makes it an ideal artifact for formally verifying the structure of a generated model. A typical file includes sections for the objective function, constraints, variable bounds, and variable types:

- **Objective Function:** Begins with `Maximize` or `Minimize`, followed by the objective function expression.
- **Constraints (`Subject To`):** This section lists all the linear constraints of the model. Each constraint is an equation or inequality involving the decision variables.
- **Bounds:** This section specifies the lower and upper bounds for each variable (e.g., `x >= 0`). If omitted, default bounds are often assumed by the solver.
- **Variable Types (`General`, `Binary`):** This section declares which variables must be integer (`General` or `Integers`) or binary (`Binary`). Variables not declared here are assumed to be continuous.
- **End:** A final `End` statement marks the end of the file.

## A.2 A Concrete Example

To illustrate the entire process, consider the following simple production-mix problem. This example shows the complete chain: from natural language to a formal model, then to Python code using a **modeling library**, which in turn uses a **solver** to produce both a final solution and a verifiable `.lp` **file**. Both the numerical solution (e.g., objective value) and the structured `.lp` artifact are crucial for providing the rich feedback signals used in our framework.

1. **Natural Language Problem:**

---

**Natural Language Problem**

A factory produces two products, tables and chairs. Each table yields a profit of $16 and requires 2 hours of labor and 4 units of wood. Each chair yields a profit of $10 and requires 1 hour of labor and 2 units of wood. The factory has 80 hours of labor and 150 units of wood available. How many tables and chairs should be produced to maximize profit?

---

2. **Mathematical Model:**

---

**Mathematical Formulation**

Let $x_{\text{tables}}$ and $x_{\text{chairs}}$ be the number of tables and chairs to produce. The formal mathematical model is:

$$
\begin{aligned}
\text{maximize} \quad & 16x_{\text{tables}} + 10x_{\text{chairs}} && \text{(Total Profit)} \\
\text{subject to} \quad & 2x_{\text{tables}} + x_{\text{chairs}} \leq 80 && \text{(Labor Constraint)} \\
& 4x_{\text{tables}} + 2x_{\text{chairs}} \leq 150 && \text{(Wood Constraint)} \\
& x_{\text{tables}}, x_{\text{chairs}} \geq 0, \in \mathbb{Z} && \text{(Integer Variables)}
\end{aligned}
$$

---

3. **Python Code:** Here is the Python code that uses the GurobiPy library to implement the mathematical model.

```
Python Code(using the GurobiPy Modeling Library)

import gurobipy as gp
from gurobipy import GRB

# Create a new model
m = gp.Model("production")

# Create variables
tables = m.addVar(vtype=GRB.INTEGER, name="tables")
chairs = m.addVar(vtype=GRB.INTEGER, name="chairs")

# Set objective
m.setObjective(16 * tables + 10 * chairs, GRB.MAXIMIZE)

# Add constraints
m.addConstr(2 * tables + 1 * chairs <= 80, "labor")
m.addConstr(4 * tables + 2 * chairs <= 150, "wood")

# Write the LP file for inspection
m.write("production_model.lp")

# Optimize model
m.optimize()

# Print solution
print(f"Objective value: {m.objVal}")
```

4. **Generated `.lp` File:**

```
production_model.lp

\ Model production
\ LP format - for model browsing.
Maximize
  16 tables + 10 chairs
Subject To
  labor: 2 tables + chairs <= 80
  wood: 4 tables + 2 chairs <= 150
Bounds
Generals
  tables
  chairs
End
```

5. **Solver Output and Solution:**
   The 'm.optimize()' command then invokes the Gurobi solver, which finds the optimal pro-
   duction plan. The typical output includes:

   - **Status:** Optimal solution found.
   - **Objective Value:** 750.0 (Maximum Profit)
   - **Decision Variables:**
     - `tables` = 0
     - `chairs` = 75

# B   Benchmark dataset

We evaluated the performance of our trained models on four key optimization modeling datasets:
NL4OPT [82], MAMO [83], IndustryOR [25], and OptMATH [26]. As noted in prior work [26, 27],

errors exist within the test sets of these benchmarks. To ensure the integrity of our evaluation, we rigorously reviewed and corrected the test sets, updating both the questions and corresponding solutions, with a specific focus on the NL4OPT and IndustryOR datasets. The sample statistics, before and after this revision process, are summarized in Table 5.

Table 5: Summary statistics for the optimization problem datasets

| Dataset | Number Before Correction | Number After Correction |
|---|---|---|
| NL4OPT | 245 | 245 |
| Mamo Easy | 652 | 642 |
| Mamo Complex | 211 | 203 |
| IndustryOR | 100 | 100 |
| OptMATH | 166 | 166 |

- **NL4OPT**: Originating from the NL4OPT competition, this benchmark assesses the conversion of natural language problem statements into executable optimization models. It predominantly features linear programming (LP) tasks drawn from diverse domains, although the complexity of the underlying mathematical structures is relatively consistent. A final set comprising 245 high-quality instances was obtained by filtering low-quality instances [26].

- **MAMO** consists of two subsets: **MAMO EasyLP** and **MAMO ComplexLP**. **MAMO EasyLP** contains 652 instances, primarily high-school level mixed-integer linear programming (MILP) problems designed for fundamental learning; **MAMO ComplexLP** includes 211 more challenging undergraduate-level problems, integrating both LP and MILP concepts suitable for assessing advanced modeling skills. Note that to guarantee the quality of this benchmark for our experiments, we performed a thorough data curation process. We manually verified and corrected the MAMO EasyLP and MAMO ComplexLP dataset.

- **IndustryOR**: Introduced as the first benchmark specifically targeting industrial applications, IndustryOR comprises 100 real-world optimization scenarios sourced from various sectors. It distinguishes itself by incorporating a wider range of problem types, including linear programming (LP), integer programming (IP), mixed-integer programming (MIP), nonlinear programming (NLP), and other specialized formulations, categorized across three difficulty levels.

- **OptMATH**: The benchmark was constructed to address the limitations of existing datasets, particularly their frequent focus on linear problems and the potentially insufficient challenge level. OptMATH features a curated selection of complex mathematical optimization problems designed to effectively differentiate advanced modeling capabilities. Its scope is intentionally broad, encompassing LP, MILP, IP, NLP, Second-Order Cone Programming (SOCP), and other relevant optimization paradigms. Two variants of the dataset exist, with 166 and 193 instances, respectively. We present the results from the larger 193-instance variant in the main results part to ensure a thorough evaluation.

Note that to ensure the validity of our findings and contribute a more robust benchmark for the community, we performed a manual verification and correction of the datasets used in our evaluation. This rigorous process involved identifying and rectifying a range of issues within the original data, including problems with contradictory or incomplete conditions, flawed logical premises, and incorrect ground-truth answers. We believe these curated versions enhance the reliability of benchmarking for future research. For the benefit of future research, a comprehensive description detailing our correction methodology, the rationale for each modification, and the revised datasets are publicly available at the Github repository https://github.com/Cardinal-Operations/SIRL/blob/main/test_data/README.md.

We evaluate performance across these benchmarks using a strict accuracy metric: a generated solution is considered correct if the related difference between its predicted objective value $y_{pred}$ and the ground truth value $y_{label}$ satisfies:

$$\frac{|y_{pred} - y_{label}|}{|y_{label}|} < 10^{-6}.$$

Similar to [26], the error about the type of the variables are ignored. We thoroughly review problems exhibiting such errors. In these cases, the decision variables typically represent time or similar quantities, which can reasonably be treated as either integer or continuous variables.

## C Seed data

The seed data consist of 686 real-world industry cases collected from operations research textbooks and a list of 100 industry scenarios [25]. Here is an example:

---

**Seed data example**

**Problem**: **Tourist Restaurant Seating Allocation Problem**
**Problem Background:**
The tourist restaurant seating allocation problem involves arranging the distribution of 100 seats in the restaurant at two different periods to improve customer satisfaction, optimize service efficiency, and maximize restaurant revenue. This is a common management challenge in the catering industry, especially during peak dining periods.
**Optimization Goals:**
- Maximize customer satisfaction: Improve customer dining experience through reasonable seat allocation.
- Maximize restaurant revenue: Maximize restaurant revenue by optimizing seat allocation.
- Optimize service efficiency: Ensure that the restaurant can operate efficiently at different times and reduce waiting time.
**Numerical Problem Constraints:**
1. **Total Seats Limit:**
- At any given period, the total number of seats allocated must not exceed the restaurant's total capacity of 100 seats.
2. **Fluctuating Demand Across Periods:**
- In Period 1, the seat allocation must be between the minimum demand of 20 seats and the maximum demand of 60 seats.
- In Period 2, the seat allocation must be between the minimum demand of 30 seats and the maximum demand of 70 seats.
**Objective Function:**
The objective is to maximize customer satisfaction and restaurant revenue, where the combined weight for customer satisfaction and revenue per allocated seat is 1.0 for each.
**Formulation:**
Let `x[i, j]` denote the allocation of seat `j` in period `i` (where `i = 1, 2` for two periods and `j = 1, 2, ..., 100` for total seats).
**Objective Function:**
Maximize:
$$ \sum_{i=1}^{2} \sum_{j=1}^{100} \left( 1.0 + 1.0 \right) x[i, j] $$
**Constraints:**
1. Total seat allocation in each period should not exceed the total available seats:
$$ \sum_{j=1}^{100} x[i, j] \leq 100, \quad \forall i \in \{1, 2\} $$
2. Seat allocation per period must satisfy minimum and maximum demand:
$$ 20 \leq \sum_{j=1}^{100} x[1, j] \leq 60 $$ $$ 30 \leq \sum_{j=1}^{100} x[2, j] \leq 70 $$
**Expected Output:**
The optimal seating allocation for each period (`x[i, j]` values) that maximizes customer satisfaction and revenue while adhering to all constraints.
**Answer**: 260.0

---

# D Prompt templates

In this section, we summarize the prompts used in the framework.

## D.1 Ten roles for self-consistency

In the self-consistency mechanism, we generate multiple candidate solutions for each problem by prompting the LLM with ten distinct roles. Each role represents a specific expert persona, designed to elicit varied approaches to optimization modeling by emphasizing different combinations of expertise in operations research, Python development, and Gurobi solver application. The ten roles are defined as:

---

**Ten roles**

- **Role 1**:"A highly skilled Python engineer and optimization specialist with deep expertise in operations research and the Gurobi solver."

- **Role 2**:"An optimization expert and Python engineer specializing in operations research and the Gurobi solver."

- **Role 3**:"A Python engineer and optimization specialist with a strong background in operations research and the Gurobi solver."

- **Role 4**:"A skilled Python engineer and optimization specialist proficient in operations research and the Gurobi solver."

- **Role 5**:"A results-driven Python engineer and optimization expert with a strong foundation in operations research and the Gurobi solver."

- **Role 6**: "A seasoned operations research scientist and Python developer, leveraging advanced optimization techniques and the Gurobi solver to tackle complex business challenges."

- **Role 7**: "An innovative optimization modeler and Python programmer, specializing in the development and implementation of high-performance solutions using operations research methodologies and the Gurobi optimization suite."

- **Role 8**: "A pragmatic problem-solver with expertise in operations research, proficient in Python and the Gurobi API, focused on translating real-world scenarios into efficient and scalable optimization models."

- **Role 9**: "A meticulous optimization analyst and Python coder, deeply familiar with the theoretical underpinnings of operations research and the practical application of the Gurobi solver for achieving optimal outcomes."

- **Role 10**: "A strategic optimization architect and Python implementation specialist, with a proven track record of designing and deploying robust operations research solutions powered by the Gurobi optimization engine."

---

## D.2 LLM-as-a-judge for generated problem evaluation

We employ the LLM-as-a-judge methodology [46] to validate the generated problems for practical relevance and semantic consistency. The prompt utilized for this validation process is detailed below.

**Problem evaluation prompt**

You are an expert in operations research. You'll receive an operations research problem. You will analyze it and determine whether the problem is a valid one considering the following aspects:
1. Determine if the problem's language and structure are consistent with typical operations research problem formulations.
2. Assess whether the problem scenario has real-world applicability or practical significance.
3. Identify any semantic inconsistencies, contradictions, or ambiguities within the problem statement.
Below is the operations research problem:
{{Question}}
Please provide your step-by-step analysis and your final judgment for each of these points.
## Analysis Process:
[Your detailed step-by-step reasoning for each point above]
## Final Judgment:
[Yes or No]

## D.3 Refine and regenerate for the error case

Upon encountering code execution errors, we leverage the LLM for refinement and code regeneration [72]. The prompt used for this error correction mechanism is as follows.

**Error regenerate prompt**

You are an experienced operations research algorithm engineer. You are presented with an operations research problem and a previous attempt to model and code a solution. That attempt resulted in an error.
Problem Description:
{{Question}}
Previous Code Solution Attempt:
{{Previous code}}
After running the provided code from the previous attempt, the following error occurred:
{{Error output after executing the code}}
Your task:
Based on the information above, please perform the following:
1. Analyze Root Cause & Identify Pitfalls
Thoroughly analyze the root cause of the error.
Summarize potential pitfalls or common mistakes related to this type of code error.
2. Provide Corrected Gurobi Code:
Write the complete and corrected Python code using the 'gurobipy' library to accurately solve the problem.
Please structure your response strictly as follows:
## Cause of the Error and Potential Pitfalls:
[Your detailed analysis of the error's cause and a summary of potential pitfalls.]
## Corrected Gurobi Code:
[Your complete and corrected Gurobi Python code.]

## D.4 Refine and regenerate for the infeasible case

If an infeasible solution is obtained after executing the code using the solver, we leverage the LLM to refine the entire result by regenerating both the mathematical model and its corresponding code [72]. The prompt used for this infeasibility resolution mechanism is as follows.

## Infeasible regenerate prompt

You are an experienced operations research algorithm engineer. You are presented with an operations research problem and a previous attempt to model and code a solution. That attempt resulted in an infeasible solution.
Problem Description:
{{Question}}
Previous Model and Code Solution Attempt:
{{Previous model}}
{{Previous code}}
After running the provided code from the previous attempt, the answer could not provide a feasible solution.
Your task:
Based on the information above, please perform the following:
1. Analyze Root Cause & Identify Pitfalls
Thoroughly analyze the root cause of the infeasibility.
Summarize potential pitfalls or common mistakes related to this type of infeasibility.
2. Provide an Improved Mathematical Model: Develop a mathematical model for correctly modeling this OR problem. This should address the flaws in the previous attempt.
3. Provide Corrected Gurobi Code:
Write the complete and corrected Python code associated with the mathematical model using the 'gurobipy' library to accurately solve the problem.
Please structure your response strictly as follows:
## Cause of the Infeasibility and Potential Pitfalls:
[Your detailed analysis of the infeasibility's cause and a summary of potential pitfalls.]
## Corrected Mathematical Model:
[Your improved mathematical model.]
## Corresponding Gurobi Code:
[Your complete and corrected Gurobi Python code associated with the mathematical model.]

## D.5 System prompt and the LLM responses

We follow the approach of DeepSeek-R1-Zero [29] by providing a specific output format within the system prompt. This format guides the LLMs through a sequential process: it first asks the LLM to analyze the problem, then formulate a mathematical model, and finally output the corresponding solver code. To ensure clear delineation and parsability of these steps, we prompt the LLM to explicitly separate each component using distinct tags: <think></think>, <model></model>, <python></python>. An example output response guided by the system prompt is provided below. The corresponding problem was sourced from the NL4OPT dataset.

## System Prompt

**SYSTEM**: You are a helpful Assistant with expertise in operations research and the Gurobi solver. When the User provides an OR question, you will analyze it, build a detailed mathematical model, and provide the Gurobi code to solve it.
Your response should follow these steps:
1. <think> Carefully analyze the problem to identify decision variables, objective, and constraints.</think>
2. <model> Develop a complete mathematical model, explicitly defining: * Sets * Parameters * Decision Variables (and their types) * Objective Function * Constraints </model>
3. <python> Provide the corresponding Gurobi Python code to implement the model. </python>
The output must be in Markdown format, with each step enclosed in the specified tags.
**USER**: Answer the following mathematical modeling question:
{{Question}}
Please think step by step.

## Problems

**Problem**: An industrial tire company delivers large tires for equipment to remote engineering sites either by cargo planes or ultrawide trucks. Each cargo plane can transport 10 tires per trip and costs $1000. Each ultrawide truck can transport 6 tires per trip and costs $700. The company needs to transport at least 200 tires and has available $22000. Because most remote sites don't have proper airports, the number of plane trips cannot exceed the number of ultrawide truck trips. How many trips of each should be done to minimize the total number of trips?

**Response**

<think> To solve this problem, we need to identify the decision variables, the objective function, and the constraints.
1. **Decision Variables:**
- Let $x$ be the number of plane trips.
- Let $y$ be the number of truck trips.
2. **Objective Function:**
We need to minimize the total number of trips, which is $x + y$.
3. **Constraints:**
- The total number of tires transported must be at least 200: $10x + 6y \geq 200$.
- The total cost must not exceed \$22000: $1000x + 700y \leq 22000$.
- The number of plane trips cannot exceed the number of truck trips: $x \leq y$.
- Non-negativity constraints: $x \geq 0$, $y \geq 0$.
Since these are linear constraints, we can use linear programming to solve this problem.
</think>
<model>
### Sets
- There are no sets in this problem as it is a simple linear programming problem.
### Parameters
- The number of tires each plane can transport: 10.
- The cost per plane trip: \$1000.
- The number of tires each truck can transport: 6.
- The cost per truck trip: \$700.
### Decision Variables
- $x$: Number of plane trips.
- $y$: Number of truck trips.
### Objective Function
Minimize the total number of trips: $x + y$.
### Constraints
1. $10x + 6y \geq 200$
2. $1000x + 700y \leq 22000$
3. $x \leq y$
4. $x \geq 0$, $y \geq 0$
</model>
<python>
```python
import gurobipy as gp
from gurobipy import GRB
# Create a new model
m = gp.Model("tire_transport")
# Decision variables
x = m.addVar(name="x", vtype=GRB.INTEGER) # Number of plane trips
y = m.addVar(name="y", vtype=GRB.INTEGER) # Number of truck trips
# Objective function
m.setObjective(x + y, GRB.MINIMIZE)
# Constraints
m.addConstr(10 * x + 6 * y >= 200, "tire_constraint")
m.addConstr(1000 * x + 700 * y <= 22000, "cost_constraint")
m.addConstr(x - y <= 0, "trip_constraint")
# Optimize the model
m.optimize()
# Print the solution
if m.status == GRB.OPTIMAL:
    print(f"Optimal solution found:")
    print(f"Plane trips: x.x")
    print(f"Truck trips: y.x")
    print(f"Total trips: x.x + y.x")
    print(f"Total cost: $m.objVal")
else:
    print("No optimal solution found.")
```
</python>

# E    Reward function design

The overall two-stage reward function $r(x, \mathbf{z}, y^*)$ is defined as follows:

$$r(x, \mathbf{z}, y^*) = \begin{cases} R_{\text{format}}(\mathbf{z}) + R_{\text{exec}}(\mathbf{z}) + R_{\text{accur}}(x, \mathbf{z}, y^*) & \text{Stage-1,} \\ R_{\text{format}}(\mathbf{z}) + R_{\text{exec}}(\mathbf{z}) + R_{\text{accur}}(x, \mathbf{z}, y^*) + R_{\text{bonus}}(x, \mathbf{z}, y^*) & \text{Stage-2.} \end{cases}$$

The reward function comprises the following components:

**Format reward** $R_{\text{format}}(\mathbf{z})$: the format reward guides the LLM policy to produce response $\mathbf{z}$ with a specific, parsable structure trajectory defined by the system prompt. This structure segregates the solution trajectory via tags like `<think></think>` for reasoning steps, `<model></model>` for the optimization model, and `<python></python>` for executable code. $R_{\text{format}}(\mathbf{z})$ is a binary reward (1 or 0) awarded only if $\mathbf{z}$ strictly includes all required tags in their correct order.

Let $\mathcal{T} = \{\texttt{<think>...</think>}, \texttt{<model>...</model>}, \texttt{<python>...</python>}\}$ be the set of required tag pairs. The reward is:

$$R_{\text{format}}(\mathbf{z}) = \begin{cases} 0.5 & \text{if } \mathbf{z} \text{ contains all tags in } \mathcal{T} \text{ according to system prompt} \\ 0 & \text{otherwise.} \end{cases}$$

This reward is also foundational for enabling the extraction and evaluation of the generated model and code.

**Execution reward** $R_{\text{exec}}(\mathbf{z})$: assigns a reward of 1 if the optimization code within response $\mathbf{z}$ is executable, and 0 otherwise.

$$R_{\text{exec}}(\mathbf{z}) = \begin{cases} 1 & \text{if the code is executable,} \\ 0 & \text{otherwise.} \end{cases}$$

**Accuracy reward** $R_{\text{accur}}(x, \mathbf{z}, y^*)$: the accuracy reward evaluates the correctness of the final answer $y = g(x, \mathbf{z})$ obtained by executing the code in $\mathbf{z}$. The answer is considered correct if matches the ground truth $y^*$ within a tolerance $|y - y^*| \leq 0.01$ . In the first stage, the reward is defined as

$$R_{\text{accur}}(x, \mathbf{z}, y^*) = \begin{cases} 2 & \text{if the answer is right,} \\ 0 & \text{otherwise.} \end{cases}$$

**Bonus accuracy reward** $R_{\text{bonus}}(x, \mathbf{z}, y^*)$: real-world optimization problems frequently involve nonlinear relationships or discrete variables, to encourage our model to tackle more complex optimization problems requiring techniques beyond standard Linear Programming (LP), we introduce a bonus reward. By analyzing the `.lp` file generated by the solver code, we can verify whether these advanced techniques (Big-M methods [84], binary variables, or nonlinear formulations) is used. The binary bonus $R_{\text{bonus}}(\mathbf{z})$ is awarded for output $\mathbf{z}$ if, and only if, both the correct answer derived from $\mathbf{z}$ is correct and the generated model utilizes advanced modeling techniques detectable through instance analysis.

$$R_{\text{bonus}}(z) = \begin{cases} 1 & \text{if advanced modeling techniques are used,} \\ 0 & \text{otherwise.} \end{cases}$$

# F    Details of experiments

**Training setup.** All experiments for the 7B model were conducted on a single compute node equipped with eight 80GB NVIDIA H100 GPUs. The two-stage training process required approximately 24 hours of wall-clock time per stage, for a total computational cost of 384 GPU hours.

Starting from the synthetic dataset, we applied a filtering strategy guided by the principle "Less is More" [73, 74]. Specifically, we excluded (question, answer) pairs if the baseline Qwen-32B-Instruct model [22] achieved an 80% success rate (8/10 attempts across different prompting roles) in generating executable code matching the ground-truth optimal value, as such samples were deemed too trivial. This process yielded approximately 70,000 samples. From this set, we then randomly sampled 10,000 instances to form our training data.

We used Qwen2.5-7B-Instruct [22] as the base model and adapted the Verl framework [85] for reinforcement learning training, modifying its implementation to incorporate our novel surrogate function design with the `Partial KL` strategy and two-stage reward mechanism.

The key hyperparameters for SIRL training are detailed in Table 6:

Table 6: Training Parameters

| Type | Parameter | Value |
|------|-----------|-------|
| **Algorithm** | Advantage Estimator | reinforce_plus_plus |
| **Data** | Batch size | 128 |
| | Learning rate | 1e-6 |
| | Max prompt length | 2048 |
| | Max response length | 8192 |
| | Truncation | left |
| **Actor/Rollout** | KL loss type | low_var_kl |
| | KL loss coefficient | 0.005 |
| | Rollout number | 8 |
| | PPO mini batch size | 8 |
| | PPO micro batch Size per GPU | 4 |
| | Clip ratio low | 0.20 |
| | Clip ratio high | 0.28 |

**Decoding strategy.** We employ the top-P (nucleus) decoding strategy [86] for the training and inference phases. The exact sampling hyperparameters used to generate our main results are specified in Table 7:

Table 7: Sampling parameters used for text generation.

| Parameter | Value |
|-----------|-------|
| n | 1 |
| Temperature | 0.5 |
| Top p | 0.9 |
| Max tokens | 8192 |
| Repetition penalty | 1.02 |

## G   Further Comparative Analysis

**Sampling-Based Performance Metrics**   To further assess the robustness of our method under stochastic generation, we conducted experiments using top-p (nucleus) sampling with temperature = 0.5 and top-p = 0.95. The mean accuracy and standard deviation across multiple runs are reported in Table 8.

Table 8: Mean pass@1 accuracy (%) $\pm$ standard deviation under top-p sampling (temperature = 0.5, top-p = 0.95).

| Metric | NL4OPT | MAMO Easy | MAMO Complex | IndustryOR | OptMATH |
|--------|--------|-----------|--------------|------------|---------|
| mean $\pm$ std | $96.2 \pm 0.32$ | $90.1 \pm 0.14$ | $52.8 \pm 1.04$ | $31.7 \pm 1.5$ | $31.4 \pm 1.29$ |

These results confirm the stability and high efficacy of our SIRL approach, even under stochastic sampling conditions, with low variance indicating consistent performance.

**Performance without External Solver Assistance**   We first evaluate the performance of models trained solely via textual step-by-step reasoning, without leveraging external solvers for verification or execution. This setup relies entirely on the LLM's internal reasoning capabilities to generate solutions. The results, reported as pass@1 accuracy, are summarized in Table 9.

Table 9: Pass@1 accuracy (%) on optimization benchmarks for models trained via textual reasoning without solvers.

| Models | NL4OPT | MAMO Easy | MAMO Complex | IndustryOR | OptMATH |
|---|---|---|---|---|---|
| Qwen2.5-7B-Instruct | 24.5 | 14.7 | 1.4 | 7.0 | 5.7 |
| Math4Opt-RL | 15.5 | 42.2 | 3.3 | 13.0 | 13.5 |

As the results in Table 9 demonstrate, this approach yields extremely low performance across most benchmarks, highlighting the limitations of LLM-only reasoning for optimization tasks. While large reasoning models (e.g., DeepSeek-R1) excel at sequential reasoning in tasks like AIME mathematics, optimization modeling introduces unique challenges. It demands a global understanding of highly interconnected objective functions, constraints, and decision variables, coupled with the execution of complex computational steps. This computational burden is difficult to manage through purely textual, step-by-step reasoning alone. In contrast, the optimization community relies on mature solvers to provide the necessary global perspective and computational efficiency. Our approach bridges this gap by using LLMs for accurate problem comprehension and initial mathematical formulation, while delegating computationally intensive aspects to specialized solvers through generated code blocks. This division of labor is central to achieving high performance in optimization tasks and motivates the integration of external tools in our framework.

**Comparison with SFT** Building on the limitations of pure textual reasoning, we next compare our online RL approach against a strong SFT baseline, where solver-assisted code generation is incorporated. The pass@1 results across models are presented in Table 10.

Table 10: Pass@1 accuracy (%) comparison: Base model, SFT, and RL.

| Models | NL4OPT | MAMO Easy | MAMO Complex | IndustryOR | OptMATH |
|---|---|---|---|---|---|
| BaseModel | 75.1 | 81.3 | 22.7 | 13.0 | 4.1 |
| SFT | 83.3 | 86.4 | 38.0 | 24.0 | 20.8 |
| RL | 96.3 | 90.0 | 51.7 | 33.0 | 30.5 |

As evidenced in Table 10, our RL method consistently outperforms both the base model and SFT across all benchmarks. The gains are especially notable on challenging datasets like MAMO Complex (24.1% over SFT) and OptMATH (8.2% over SFT), emphasizing RL's role in enhancing complex problem-solving through iterative feedback and solver verification. This progression from pure reasoning to hybrid RL-solver paradigms illustrates a scalable path toward robust optimization modeling with LLMs.

**Error Type Analysis** To dissect the sources of failure and provide deeper insights into the improvements, we categorize errors into types such as code extraction failures, timeouts, execution errors, and wrong answers (with the remainder being correct solutions). Detailed breakdowns are provided in Tables 11 and 12 for the base Qwen-2.5-7B-Instruct model and our SIRL-Qwen2.5-7B variant, respectively.

Table 11: Error type distribution (%) for Qwen-2.5-7B-Instruct.

| Error Type | NL4OPT | MAMO Easy | MAMO Complex | IndustryOR | OptMATH |
|---|---|---|---|---|---|
| Code Extraction Failed | 0 | 0 | 0.4 | 0 | 2.1 |
| Timeout | 0 | 0 | 0 | 0 | 0.5 |
| Execution Error | 6.1 | 3.2 | 33.6 | 30 | 73.1 |
| Wrong Answer | 18.0 | 15.5 | 43.2 | 67 | 20.2 |
| Correct | 75.1 | 81.3 | 22.7 | 13 | 4.1 |

Comparing the two tables, our SIRL method substantially reduces execution errors and wrong answers, particularly on simpler benchmarks like NL4OPT and MAMO Easy, where correct rates

Table 12: Error type distribution (%) for SIRL-Qwen2.5-7B.

| Error Type | NL4OPT | MAMO Easy | MAMO Complex | IndustryOR | OptMATH |
|---|---|---|---|---|---|
| Code Extraction Failed | 0 | 0 | 0 | 0 | 1.6 |
| Timeout | 0 | 0 | 0 | 0 | 0 |
| Execution Error | 0.4 | 0 | 1.9 | 4 | 11.9 |
| Wrong Answer | 3.3 | 10.3 | 36.5 | 61 | 56.0 |
| Correct | 96.3 | 89.7 | 63.6 | 35 | 30.6 |

exceed 89%. On more complex datasets (e.g., MAMO Complex and OptMATH-193), execution errors persist but are markedly lower than in the base model, underscoring the value of reinforcement learning in refining code generation and solver integration.

## H  In-depth analysis of the `Partial KL` strategy

**Full ablation study on different surrogate function designs.**  Here, we present a detailed analysis of the Partial KL divergence. Table 13 shows the results of an ablation study on all benchmarks, which, due to page limitations, was not included in the main paper. The results are consistent with those reported in the original manuscript, the surrogate function design employing `Without KL` strategy demonstrates a significantly reduced execution rate compared to the other two designs.

Table 13: Ablation study on Partial KL

| Type | NL4OPT | | MAMOEasy | | MAMOComplex | | IndustryOR | | OptMATH | |
|---|---|---|---|---|---|---|---|---|---|---|
| | Acc(pass@1) | ER | Acc(pass@1) | ER | Acc(pass@1) | ER | Acc(pass@1) | ER | Acc(pass@1) | ER |
| Partial KL | 96.3% | 100.0% | 91.7% | 100.0% | 51.7% | 98.1% | 33.0% | 96.0% | 30.5% | 92.2% |
| Full KL | 95.1%(↓1.2%) | 99.2% | 89.9%(↓0.1%) | 99.7% | 48.3%(↓3.4%) | 97.6% | 30%(↓3.0%) | 95.0% | 28.3%(↓2.2%) | 93.4% |
| Without KL | 92.7%(↓3.6%) | 98.5% | 88.7%(↓1.3%) | 100.0% | 47.3%(↓4.4%) | 95.6% | 29%(↓4.0%) | 87.0% | 29.5%(↓1.0%) | 80.1% |

**Case study on different surrogate function designs.**  In this part, two case studies are presented to demonstrate how different surrogate function configurations affect LLM-generated responses. The first case study demonstrates the limitations of `Without KL` strategies, showcasing execution errors in an elementary mathematical modeling question; The second examines how the `Partial KL` strategy, by selectively removing the KL divergence term, enhances reasoning capabilities to better understand questions, especially those involving logic constraints.

**Case Study 1:**  The first example is chosen from the IndustryOR dataset. In this case, we compare the generated code snippets to present the execution error arising from the model trained with the `Without KL` strategy. We observe that omitting the KL divergence in the code component leads to execution errors. Specifically, as marked in red, the response from the `Without KL` model attempts to enforce that decision variables are greater than zero, but the implemented code is non-functional since '>' not supported between instances of 'Var' and 'int'.

**Case study 2:**  The second case study involves a problem from the IndustryOR dataset, characterized by more complex logical constraints. In this case, an incorrect reasoning step (highlighted in red) leads to a flawed problem formulation. This category of problems demands a deep understanding of the underlying structure and inherent logical dependencies. We observe that omitting the KL divergence penalty encourages the model to enhance its reasoning capabilities. This fosters more effective exploration of the solution space.

---

**Case Study 1: Question instance from the IndustryOR dataset**

A product can be processed on any one of the four devices: A, B, C, or D. The preparation completion costs when each device is enabled, the unit production cost for the product, and the maximum processing capacity of each device are shown in Table 5-7. If 2000 units of the product need to be produced, how can the total cost be minimized? Try to establish a mathematical model.
Table 5-7 Device,Prep Completion Cost (Yuan),Unit Production Cost (Yuan/Unit),Maximum Processing Capacity (Units) A,1000,20,900 B,920,24,1000 C,800,16,1200 D,700,28,1600

---

## Case Study 1: Generated response under the `Full KL` strategy

```python
<python>
import gurobipy as gp
from gurobipy import GRB
# Create a new model
m = gp.Model("product_production")
# Decision variables
x_A = m.addVar(vtype=GRB.CONTINUOUS, name="x_A")
x_B = m.addVar(vtype=GRB.CONTINUOUS, name="x_B")
x_C = m.addVar(vtype=GRB.CONTINUOUS, name="x_C")
x_D = m.addVar(vtype=GRB.CONTINUOUS, name="x_D")
y_A = m.addVar(vtype=GRB.BINARY, name="y_A")
y_B = m.addVar(vtype=GRB.BINARY, name="y_B")
y_C = m.addVar(vtype=GRB.BINARY, name="y_C")
y_D = m.addVar(vtype=GRB.BINARY, name="y_D")
# Objective function
m.setObjective(1000 * y_A + 920 * y_B + 800 * y_C + 700 * y_D + 20 * x_A + 24 * x_B + 16 * x_C
+ 28 * x_D, GRB.MINIMIZE)
# Constraints
m.addConstr(x_A + x_B + x_C + x_D == 2000, "total_production")
m.addConstr(x_A <= 900 * y_A, "capacity_A")
m.addConstr(x_B <= 1000 * y_B, "capacity_B")
m.addConstr(x_C <= 1200 * y_C, "capacity_C")
m.addConstr(x_D <= 1600 * y_D, "capacity_D")
# Optimize the model
m.optimize()
# Print the solution
if m.status == GRB.OPTIMAL:
print(f"Total cost: {m.objVal}")
print(f"Units processed on A: {x_A.x}")
print(f"Units processed on B: {x_B.x}")
print(f"Units processed on C: {x_C.x}")
print(f"Units processed on D: {x_D.x}")
print(f"Device A enabled: {y_A.x}")
print(f"Device B enabled: {y_B.x}")
print(f"Device C enabled: {y_C.x}")
print(f"Device D enabled: {y_D.x}")
else:
print("No optimal solution found.")
</python>
```

## Case Study 1: Generated response under the `Without KL` strategy

```python
<python>
from gurobipy import *
# Define the model
model = Model("Product_Optimization")
# Parameters
prep_completion_cost = {"A": 1000, "B": 920, "C": 800, "D": 700}
unit_production_cost = {"A": 20, "B": 24, "C": 16, "D": 28}
max_processing_capacity = {"A": 900, "B": 1000, "C": 1200, "D": 1600}
# Decision variables
x = {}
for i in prep_completion_cost.keys():
x[i] = model.addVar(name=i, lb=0, ub=max_processing_capacity[i], vtype=GRB.CONTINUOUS)
# Objective function
model.setObjective(quicksum(prep_completion_cost[i] * (x[i] > 0) + unit_production_cost[i] * x[i] for
i in prep_completion_cost.keys()), GRB.MINIMIZE)
# Constraints
model.addConstr(quicksum(x[i] for i in prep_completion_cost.keys()) == 2000, "Total_Production")
# Optimize the model
model.optimize()
# Print the solution
if model.status == GRB.Status.OPTIMAL:
print("Optimal Solution:")
for i in prep_completion_cost.keys():
print(f"Device {i}: {x[i].x}")
print(f"Total Cost: {model.objVal}")
else:
print("No optimal solution found.")
</python>
```

**Case Study 1: Generated response under the `Partial KL` strategy**

```python
<python>
from gurobipy import *
# Define the parameters
prep_completion_cost = {'A': 1000, 'B': 920, 'C': 800, 'D': 700}
unit_production_cost = {'A': 20, 'B': 24, 'C': 16, 'D': 28}
max_processing_capacity = {'A': 900, 'B': 1000, 'C': 1200, 'D': 1600}
# Create a model
model = Model("ProductProduction")
# Define decision variables
x = {}
y = {}
for device in max_processing_capacity.keys():
    x[device] = model.addVar(vtype=GRB.BINARY, name=f"x_{device}")
    y[device] = model.addVar(name=f"y_{device}")
# Set objective
model.setObjective(quicksum(prep_completion_cost[device] * x[device] +
unit_production_cost[device] * y[device] for device in max_processing_capacity.keys()),
GRB.MINIMIZE)
# Add constraints
model.addConstr(quicksum(y[device] for device in max_processing_capacity.keys()) == 2000)
for device in max_processing_capacity.keys():
    model.addConstr(y[device] <= max_processing_capacity[device] * x[device])
# Optimize the model
model.optimize()
# Print the optimal solution
if model.status == GRB.OPTIMAL:
    print("Optimal solution found:")
    for device in max_processing_capacity.keys():
        print(f"Device {device}: x = {x[device].x}, y = {y[device].x}")
    print(f"Minimum cost: {model.objVal}")
else:
    print("No optimal solution found.")
</python>
```

**Case Study 2: Question instance from the IndustryOR dataset**

A manufacturing company needs to transport 1800 units of product from the warehouse to three different sales points. The company has four transportation options to choose from: truck, van, motorcycle, and electric vehicle. Since the van and electric vehicle both consume a lot of energy, the company wants to choose only one of these two options. Each trip with a truck generates 100 units of pollution, a van generates 50 units of pollution, a motorcycle generates 10 units of pollution, and an electric vehicle generates 0 units of pollution. The total pollution generated from all trips cannot exceed 2000 units. At least 10 trips must use a truck. Trucks, vans, motorcycles, and electric vehicles can transport 100 units, 80 units, 40 units, and 60 units of product per trip, respectively. The company needs to ensure that the total amount of transported product is at least 1800 units.

## Case Study 2: Generated response under the `Full KL` strategy

<think>
To solve this problem, we need to identify the decision variables, the objective function, and the constraints.
1. **Decision Variables**:
- Let $T$ be the number of trips using trucks.
- Let $V$ be the number of trips using vans.
- Let $M$ be the number of trips using motorcycles.
- Let $E$ be the number of trips using electric vehicles.
2. **Objective Function**:
- The objective is to minimize the total pollution, which can be expressed as the equation $100T + 50V + 10M + 0E$.
3. **Constraints**:
- The total product transported must be at least 1800 units: $100T + 80V + 40M + 60E \geq 1800$.
- The total pollution generated cannot exceed 2000 units: $100T + 50V + 10M \leq 2000$.
- At least 10 trips must use trucks: $T \geq 10$.
- Non-negativity constraints: $T, V, M, E \geq 0$.
4. **Binary Variable for Van or Electric Vehicle**:
- Since the company wants to choose only one of the van or electric vehicle, we can use a binary variable $B$ where $B = 1$ if a van is used and $B = 0$ if an electric vehicle is used. This can be handled by the constraints $V \leq B$ and $E \leq 1 - B$.
Let's now develop the mathematical model and provide the Gurobi Python code.
</think>
<model>
### Sets
- No sets are needed for this problem.
### Parameters
- Pollution per trip for trucks: 100
- Pollution per trip for vans: 50
- Pollution per trip for motorcycles: 10
- Pollution per trip for electric vehicles: 0
### Decision Variables
- $T$: Number of trips using trucks
- $V$: Number of trips using vans
- $M$: Number of trips using motorcycles
- $E$: Number of trips using electric vehicles
- $B$: Binary variable (1 if van is used, 0 if electric vehicle is used)
### Objective Function
- Minimize total pollution: $\text{Minimize } 100T + 50V + 10M$
### Constraints
1. Total product transported: $100T + 80V + 40M + 60E \geq 1800$
2. Total pollution: $100T + 50V + 10M \leq 2000$
3. At least 10 trips using trucks: $T \geq 10$
4. Binary variable constraint for van or electric vehicle: $V \leq B$ and $E \leq 1 - B$
5. Non-negativity constraints: $T, V, M, E, B \geq 0$
</model>

## Case Study 2: Generated response under the `Without KL` strategy

<think>
To solve this problem, we need to identify the decision variables, the objective function, and the constraints.
1. **Decision Variables:**
- Let $T$ be the number of trips made by trucks.
- Let $V$ be the number of trips made by vans.
- Let $M$ be the number of trips made by motorcycles.
- Let $E$ be the number of trips made by electric vehicles.
2. **Objective Function:**
The objective is to minimize the total pollution generated, which can be expressed as:
$$\text{Minimize } 100T + 50V + 10M$$
3. **Constraints:**
- The total amount of product transported must be at least 1800 units: $$100T + 80V + 40M + 60E \geq 1800$$
- The total pollution generated cannot exceed 2000 units: $$100T + 50V + 10M \leq 2000$$
- At least 10 trips must use a truck: $$T \geq 10$$
- Non-negativity constraints: $$T, V, M, E \geq 0 \quad \text{(integer values)}$$
Since the problem involves integer values for the decision variables, we will use integer linear programming (ILP) to solve it.
</think>
<model>
### Sets
- There are no sets in this problem as it is a simple linear programming problem.
### Parameters
- The product to be transported: 1800 units.
- Pollution generated by each type of vehicle: truck (100), van (50), motorcycle (10), electric vehicle (0).
- Product transported by each type of vehicle: truck (100), van (80), motorcycle (40), electric vehicle (60).
- Maximum pollution: 2000 units.
- Minimum truck trips: 10.
### Decision Variables
- $T$: Number of trips made by trucks.
- $V$: Number of trips made by vans.
- $M$: Number of trips made by motorcycles.
- $E$: Number of trips made by electric vehicles.
### Objective Function
$$\text{Minimize } 100T + 50V + 10M$$
### Constraints
1. Product transportation constraint: $$100T + 80V + 40M + 60E \geq 1800$$
2. Pollution constraint: $$100T + 50V + 10M \leq 2000$$
3. Minimum truck trips: $$T \geq 10$$
4. Non-negativity constraints: $$T, V, M, E \geq 0 \quad \text{(integer values)}$$
</model>

## Case Study 2: Generated response under the `Partial KL` strategy

<think>
To solve this problem, we need to define the decision variables, objective function, and constraints based on the given transportation options and pollution limits. Here's a step-by-step approach:
1. **Decision Variables:**
- Let $x_1$ be the number of trips with trucks.
- Let $x_2$ be the number of trips with vans.
- Let $x_3$ be the number of trips with motorcycles.
- Let $x_4$ be the number of trips with electric vehicles.
- Let $y$ be a binary variable where $y = 1$ if a van is chosen, and $y = 0$ if an electric vehicle is chosen.
2. **Objective Function:**
- The objective is to minimize the total pollution generated. The pollution generated by trucks is 100 units per trip, by vans is 50 units per trip, by motorcycles is 10 units per trip, and by electric vehicles is 0 units per trip.
- Therefore, the objective function is:
$$\text{Minimize } 100x_1 + 50x_2 + 10x_3 + 0x_4$$
3. **Constraints:**
- The total product transported must be at least 1800 units.
- The total pollution generated must be at most 2000 units.
- At least 10 trips must use a truck.
- Only one of the van or electric vehicle can be chosen.
</think>
<model>
### Sets
- $I = \{1, 2, 3, 4\}$ where $I$ is the set of transportation options.
### Parameters
- $P_i$ is the pollution generated by transportation option $i$.
- $C_i$ is the capacity of transportation option $i$.
### Decision Variables
- $x_1$ is the number of trips with trucks.
- $x_2$ is the number of trips with vans.
- $x_3$ is the number of trips with motorcycles.
- $x_4$ is the number of trips with electric vehicles.
- $y$ is a binary variable where $y = 1$ if a van is chosen, and $y = 0$ if an electric vehicle is chosen.
### Objective Function
$$\text{Minimize } 100x_1 + 50x_2 + 10x_3 + 0x_4$$
### Constraints
$$100x_1 + 80x_2 + 40x_3 + 60x_4 \geq 1800 \quad \text{(Total product constraint)}$$
$$100x_1 + 50x_2 + 10x_3 + 0x_4 \leq 2000 \quad \text{(Total pollution constraint)}$$
$$x_1 \geq 10 \quad \text{(At least 10 trips with trucks)}$$
$$y \in \{0, 1\} \quad \text{(Binary variable for van or electric vehicle)}$$
$$x_2 \leq 100y \quad \text{(Van constraint)}$$
$$x_4 \leq 100(1 - y) \quad \text{(Electric vehicle constraint)}$$
</model>

