# OpenReview forum: "Solver-Informed RL: Grounding Large Language Models for Authentic Optimization Modeling"
_NeurIPS.cc/2025/Conference — NeurIPS 2025 poster_

### Official Review · Reviewer_DwV1 · 2025-06-19

**Clarity:** 3
**Significance:** 2
**Originality:** 2
**Rating:** 4
**Confidence:** 3

**Summary:**

The paper proposes a reinforcement learning with verifiable rewards training pipeline to improve LLMs' capabilities to interface with external optimization solvers. Given verbal descriptions of optimization problems, the language model needs to write down the corresponding mathematical formula, variables and constraints, as well as code that can be executed by the solver to find the optimal solution. Specifically, the work proposes a synthetic training data generation pipeline to derive ground truth answers, using a per-feature self-consistency score, and designs a reward function for this task, balancing answer format, code executability, answer correctness and the use of advanced modelling techniques. In RL training, the paper proposes to maintain the KL penalty in Reinforce++ for the mathematical and code answers but remove it for the verbal reasoning parts. The proposed modifications improve performance on several evaluation benchmarks compared to ablations and prior baselines.

**Questions:**

- Did you consider other ways of encouraging exploration in verbal reasoning other than removing the KL penalty, such as using a higher temperature at training time? Or other ways of maintaining the correct format for math or code, such as structured output / limiting the token vocabulary when appropriate?

- Why does inst_sc outperform val_sc? What kind of omissions or mistakes do the highest-performing solution attempts chosen by val_sc make? How did you identify inst_sc was an appropriate strategy for those issues?

- How is the two-stage RL training implemented: when are stage 1 rewards replaced by stage 2 rewards?

**Ethical Concerns:**

["NO or VERY MINOR ethics concerns only"]

**Final Justification:**

The reviewers have adequately addressed all of the questions and the weaknesses 3-5 that I raised.

The answers to question 2 and weakness 3, in particular, improve the paper, and clarify the value and motivation of instance-enhanced self-consistency.

However, I will not increase my score to Accept (5) due to modest conceptual novelty, as highlighted in weaknesses 1 and 2. Still, the contributions, though small, seem valid and empirically justified, which is why I increased to Borderline Accept (4).

**Limitations:**

Limitations of the work are not discussed, despite claiming so in the NeurIPS checklist.

**Quality:**

2

**Strengths And Weaknesses:**

**Strengths**:
- As a fully simulated environment, defining specifications for optimization problem solvers based on the achieved objective value is a scalable source of ground truth rewards for RL training, and therefore a good application domain for reasoning model training strategies.
- Empirical results are encouraging: fine-tuned Qwen2.5-7B outperforms advanced models GPT-4 and DeepSeek-V3 as well as 5 previous LLM-for-optimization methods.
- Ablations validate the contribution of each proposed component.

**Weaknesses**:
- The main weakness is the work’s limited novelty. The novel parts, namely the selective application of the KL-penalty to certain CoT segments, and computing self-consistency on a per-dimension basis, are rather minor changes to existing RLVR approaches, especially given that the KL-penalty is already commonly omitted in RLVR training for mathematical problems in previous works.
- The presentation and writing could be improved. As it stands, the main contributions come across as empirical and out of the blue, without much general motivation presented. A thorough motivating analysis for why and how they were derived and why they are a sensible solution to specific observed issues would help in understanding the generalizability of the findings. Even if the implementation changes themselves are small, it could make for a stronger paper if they were well motivated.
- The reported results do not include error bars, despite claiming so in the NeurIPS checklist, so the statistical significance of the results is unclear.
- The problem description in the abstract and introduction is very generic (e.g. optimization "across diverse domains"): a motivating example of the problem to be solved would improve readability for a non-OR audience. Some of the description at the beginning of Section 3.2 could be mentioned earlier.
- Some experimental details are not clarified, such as the meaning and implementation of the 10 LLM roles. The relevant section in the main text should refer to the corresponding Appendix section. It is also not explained when stage 1 rewards are replaced with stage 2 rewards.

---

> ### Author Rebuttal · Authors · 2025-07-29
>
> **Question 1: Other ways of encouraging exploration and approaches to maintain the correct format for math or code**
>
> **A-1: Other ways of encouraging exploration**
>
> Thank you for this insightful question. Indeed, we did experiments with several alternative sampling strategies.
> we tested various temperature settings (e.g., T=0.8, 1.0, 1.2) with top-p sampling and also explored more recent methods like min-P[1].
> Empirically, the default configuration of top-p sampling(T=1.0, top-p=0.95) paired with our method yielded consistently good performance.
>
> Reference list:
>
>    [1]. Nguyen, M. N., Baker, A., Neo, C., Roush, A., Kirsch, A., & Shwartz-Ziv, R. (2024). Turning up the heat: Min-p sampling for creative and coherent llm outputs
>
>
> **A-2: maintenance of correct format for math or code**
>
>  Compared to fixed structured output or limiting token vocabulary, our primary mechanism is to integrate system prompt template and the precisely aligned reward mechanism.
>
> Inspired by DeepSeek-R1's practices, we use the system prompt o guide the overall output format.
> Correspondingly, we explicitly define a ``format reward'' derived from successfully parsing specific delimiters such as <model>...</model> and <python>...</python>, alongside an ```execution reward'' for runnable code.
>  This design directly encourages the policy to maintain and improve the correct format through active learning and feedback.
>
> This reward-based approach, combined with our selective KL penalty for policy control, provides a more general and adaptable solution for LLMs in optimization modeling, accommodating the diverse programming language styles of various solvers, e.g., Gurobi, Cplex, COPT.
>
> ---
>
> **Question 2: Effectiveness of Val_sc**
>
> **Response:**
>
>  Thank you for this insightful question. The key reason "inst_sc" outperforms "val_sc" is that it moves beyond comparing only the final objective value. "Val_sc" is a "black-box" method that can be misled when multiple, structurally different (and often incorrect) model formulations happen to produce the same numerical answer by coincidence. "Inst_sc" provides a more discerning evaluation by enforcing consensus on the underlying mathematical structure itself, using features extracted from the solver's LP file.
>
> A primary mistake made by top-scoring solutions chosen by "val_sc" is incorrect variable types. LLMs often struggle with the semantic nuance between a binary choice (e.g., "to build a facility or not," requiring a binary variable) and a general discrete quantity (requiring an integer variable). A solution might incorrectly use an integer variable, but if the optimal value happens to be 0 or 1, "val_sc" would incorrectly validate it based on the objective value. Other errors include using the wrong optimization direction (e.g., maximization instead of minimization).
>
> We identified "inst_sc" as the appropriate strategy after observing these specific failure modes. Since the LP file provides objective, structured statistics of the model, we hypothesized that incorporating its structural features (like binary/integer variable counts and optimization direction) into the consensus check would directly target these common LLM errors. Our experiments confirmed this, showing "inst_sc" effectively filters out these structurally flawed models and improves accuracy.
>
> ---
>
> **Question 3: Two-stage RL Training Implementation**
>
> **Response:**
>
> The training process involves two distinct stages, both utilizing the exact same training dataset.
>    * In stage 1, we utilize the specific 'stage-1 reward' signal to guide the training.
>    * For stage 2, training commences from the checkpoint obtained at the conclusion of Stage 1.
>    * Both stages  required approximately 24×8 H100 GPU hours for completion, totally 48×8 H100 hours.
>
> ---
>
> **Weakness1 and Weakness2:  Motivation and Limited Novelty**
>
> **Response:**
>
> Thank you for your constructive and essential feedback. We appreciate your perspective on the novelty and motivation of our work. We want to clarify that our core contribution lies not in proposing entirely new RLVR mechanisms in isolation, but in critically adapting and refining existing RLVR techniques to overcome fundamental limitations of LLMs when tackling complex optimization problems that necessitate external solver interaction.
>
> While LRMs (e.g., DeepSeek-R1) excel at sequential reasoning for tasks like AIME mathematics, optimization modeling presents a unique challenge. It inherently demands a global understanding of highly interconnected objective functions, constraints, and decision variables. Furthermore, it requires complex computational steps (e.g., executing the simplex method for linear programming, branch-and-bound for mixed-integer problems), which can be difficult to handle through purely textual, step-by-step reasoning.
>
> Below are early results using purely textual step-by-step reasoning, without external solver assistance:
>
> | Models                | NL4OPT | MAMO Easy | MAMO Complex | IndustryOR | OptMATH-193 |
> |-----------------------|--------|-----------|--------------|------------|-------------|
> | Qwen2.5-7B-Instruct   | 24.5%  | 14.7%     | 1.4%         | 7.0%       | 5.7%        |
> | Math4Opt-RL           | 15.5%  | 42.2%     | 3.3%         | 13.0%      | 13.5%       |
>
> As these results demonstrate, relying solely on an LLM's sequential reasoning leads to extremely low performance across all benchmarks. The established practice within the optimization community is to utilize mature solvers precisely because they offer the global view and computational power necessary to resolve these intricate interdependencies.
>
> By leveraging LLMs for accurate problem understanding and initial mathematical formulation, we can delegate the computationally intensive aspects of optimization to robust, specialized solvers through the generation of solver code blocks. This crucial division of labor is fundamental to achieving high performance in optimization tasks.
>
> Building on these findings, one of the main challenges  is to achieve both diverse reasoning exploration (which LLMs are good at) and strict accuracy and validity (which solvers demand).
> The selective KL-penalty with the tailed solver-based reward design, offers a robust and generalizable solution, makes it highly adaptable for extending LLM assistance across a variety of optimization solvers.
>
> ---
>
> **Weakness3: Error Bars of the Results**
>
> **Response:**
>
>  We sincerely apologize for the oversight in not including this crucial information in our initial submission. We have now conducted ten runs using top-p (Nucleus) sampling (temperature=0.5, top-p=0.95). Here are the mean and standard deviation results:
>
> | Metric         | NL4OPT           | MAMO Easy        | MAMO Complex     | IndustryOR      | OptMATH-193      |
> |:-------------- |:---------------- |:---------------- |:---------------- |:--------------- |:---------------- |
> | mean ± std     | 96.2% ± 0.32%    | 90.1% ± 0.14%    | 61.3% ± 0.68%    | 33.7% ± 1.5%    | 28.3% ± 0.66%    |
>
> As the table shows, our results are stable and consistent with those presented in the initial submission.
>
> ---
>
> **Weakness4: Writing of the Problem Description**
>
> **Response:**
>
> Thank you for this very constructive feedback. We fully agree that a concrete, motivating example would significantly improve the manuscript's readability and accessibility, especially for readers without a background in Operations Research. Your suggestions are extremely helpful for strengthening the introduction.
>
> In our revision, we will restructure the introduction to better frame the core challenge. We will incorporate a classic motivating example—such as a logistics or resource allocation problem—to illustrate the practical journey from a real-world business scenario to a formal mathematical model. This will help ground the generic phrase "optimization across diverse domains" by providing a tangible application.
>
> Furthermore, as you astutely pointed out, the description of the modeling process at the beginning of Section 3.2 is better suited for an earlier section. We will move this high-level overview—which describes the stages of problem analysis, mathematical formulation, and code implementation—into the introduction. This will provide readers with a clear, upfront understanding of the complex task our work aims to automate.
>
> We believe these changes will make the problem statement clearer from the outset and the motivation for our SIRL framework more compelling to a broader audience. Thank you again for your valuable guidance.
>
> ---
>
> **Weakness5: Experimental Details Clarification**
>
> **Response:**
>
> We acknowledge that the role and implementation of the "10 LLM roles" were not sufficiently detailed in the main text, and we appreciate the opportunity to clarify.
>
> The primary rationale for using 10 distinct LLM roles is to enhance the diversity of the generated outputs. By prompting the model with different expert personas (e.g., an "operations research scientist" versus a "Python engineer"), we encourage it to explore varied reasoning paths and formulation strategies for the same problem. This diversity is crucial for the robustness of any self-consistency method; it mitigates the risk of all generation attempts converging on the same plausible but incorrect solution due to a shared systemic bias in the model. A more diverse pool of candidate solutions increases the probability that the correct formulation is present and can be identified by our consensus mechanism.
>
> Due to page limitations, we placed the specific prompts that define each of these roles in the Appendix. We recognize our oversight in not explicitly linking to this from the main text. In our revised manuscript, we will amend Section 3.1 ("Instance-enhanced self-consistency") to briefly explain the purpose of these roles and add a direct reference to Appendix C.1 for the detailed prompts.
>
> ---

---

> > ### Comment · Reviewer_DwV1 · 2025-08-05
> >
> > Thank you for the thorough answers. The questions 1-3 and the weaknesses 3-5 have been addressed.
> >
> > The answers to question 2 and weakness 3 do improve the paper in my view, and clarify the value of instance-enhanced self-consistency. However, weaknesses 1 and 2 remain.
> >
> > While the contributions are relatively minor, they could be impactful in practice. I will raise my score to 4.

---

> > > ### Author Response · Authors · 2025-08-07
> > >
> > > Thank you very much for your detailed feedback throughout the review process.
> > >
> > > We are pleased that our clarifications helped address some of your initial concerns and were seen as an improvement to the paper.We also take note of your remaining concerns. We are very encouraged by your final assessment that our contributions could be impactful in practice.
> > >
> > > Thank you once again for your constructive evaluation and support.

---

### Official Review · Reviewer_fAF4 · 2025-06-25

**Clarity:** 3
**Significance:** 2
**Originality:** 2
**Rating:** 3
**Confidence:** 3

**Summary:**

This paper presents SIRL, a learning framework that finetunes LLMs for optimization modeling using reinforcement learning. Using optimization solver as reward verifiers and with a partial KL surrogate reward design, the proposed method achieves performance improvement on many optimization modeling benchmarks.

**Questions:**

See Weaknesses

**Ethical Concerns:**

["NO or VERY MINOR ethics concerns only"]

**Final Justification:**

My main concern for this paper is that the test benchmarks evaluated by the methods are often times faulty. In my experience, there could be up to 20-30% of the data in the test benchmarks that are wrong (e.g. all TSP in MAMO complex has wrong "ground truth" answers, and the IndustryOR benchmark has gone through multiple iterations of ground truth value correction up till last month, and many instances in Optmath are ambiguous). Given that the methodology proposed in this paper is purely empirical, I feel like a thorough cleaning of the test benchmarks is necessary to justify the performance improvement from the proposed method. Another concern is that, I understand this paper is the first paper that uses RL to train a LLM for optimization formulation, but given the abundance of paper that uses RL to train LLMs for math / coding / planning tasks, I do not feel like the paper is novel enough.

Notes regarding empirical rigor: if the AC reads the authors' response carefully, we can see a significant performance drop of their method after they clean up the wrong TSP labels in MAMO complex, while the performance of GPT-o3 and Deepseek-r1 significantly improved -- in my experience, there are many other wrong instances in MAMO complex beyond TSPs, and I'm worried that their model's performance will further drop  after those wrong labels are fixed. Furthermore, I feel like the authors should have taken the initiative to clean up the test benchmarks before the paper submission, instead of relying on the reviewer to tell them where in the benchmark is wrong. I personally feel like the way that the authors ask the reviewer for the cleaned test dataset feels super unjustified given they should be responsible for their results.

**Limitations:**

While the authors claim in the checklist they discuss limitations in the conclusion section, I do not see much discussion on the limitation in the conclusion. Hence, I do not believe the authors adequately addressed the limitations of the work. The authors should provide a more open discussion in terms of the limitations of the work.

**Quality:**

3

**Strengths And Weaknesses:**

[Strengths]

1. The paper is clear and easy-to-follow.
2. I find the partial KL design interesting and makes sense in the RL training context. The reward bonus design in sec 3.3 also makes sense.
3. There seems to be reasonable performance improvement comparing with the baseline, if we look at the numbers in the result table. (However, I have doubts in the comparsion, so please see my comments in weakness 4 for details.)

[Weakness]

1. While the authors mention they will provide the checkpoint and test code in the appendix, it is unclear if they will open source the training data and the training code. Hence, I feel it’s questionable regarding the commitment of the open source. I wonder if the authors can comment on their plan to open source the training data and the full code (including training)?
2. Limited contribution of the training data: many previous works propose to synthesize training data (e.g. ORLM, LLMOPT, OptMath). I’m a bit confused why the authors in this paper proposes another data synthesis framework. What’s the difference between the proposed data synthesis framework and those in the previous works? Why not just use the previous papers’ synthesized datasets? The contribution in terms of the training dataset seems limited given its very similar to those proposed in previous works.
3. Novelty may be a little limited in terms of RL training: RL has been widely used in post-training LLMs for many domains such as math / competitive programming. So, one can argue that the contribution of RL training proposed in this work is limited.
4. Questionable baseline comparisons:
     - I find the comparison with LLMOpt and OptMath a bit questionable. I believe the LLMOpt paper reports a higher performance via self-correction and OptMath has better results with their 14b and 32b models. The authors should report the best of those baselines instead of the weakest when comparing the results.
    - Furthermore, the authors should also report the performance of the SOTA reasoning models (E.g. GPT-o3, Deepseek-r1, instead of only reporting the non-reasoning models).
    - Lastly, the authors should also compare the RL model with (1) pure supervised-finetuning on their synthesized training data and (2) combining supervised-finetuning with RL to see the benefit of RL training (if any).

---

> ### Author Rebuttal · Authors · 2025-07-29
>
> **Weakness 1: Open Source Commitment**
>
> **Response:**
>
> We appreciate the reviewer’s question regarding our open-source commitment. We  confirm our plan to open-source the full training codes and a significant portion of training data.
>
>    * To demonstrate this commitment, we have already released two checkpoints (SIRL-Qwen2.5-7B-Gurobi and SIRL-Qwen2.5-7B-COPT) for immediate community use, and we plan to release more.
>    * In addition, we are actively preparing to release the complete training codebase and a significant portion of our training dataset. Our goal is to empower the community to build their own frameworks for diverse optimization problems or different optimization tools based on our work.
>
> ---
>
> **Weakness 2: Contribution of the Training Data**
>
> **Response:**
>
> Thank you for this insightful question. We acknowledge that our data synthesis pipeline builds upon foundational works like ORLM. Our primary motivation stems from the distinct data requirements of our Reinforcement Learning (RL) approach compared to traditional Supervised Fine-Tuning (SFT). While SFT relies on (question, CoTs, code) style solution trajectories, our RL framework requires high-quality (question, answer) pairs to generate a reliable reward signal.
>
> Therefore, our contribution is not a completely new synthesis framework, but a crucial enhancement focused on maximizing the correctness of the final answer. We achieve this with new emerging techniques for the LLM community(LLMs as a judge, reflection) and also our novel instance-enhanced self-consistency ("inst_sc") method. The instance-enhanced self-consistency goes beyond traditional majority voting on the final value by leveraging the solver's LP file to enforce consensus on the underlying mathematical structure (e.g., variable types). As shown in our experiments, this provides more robust verification, ensuring the high-quality data essential for our RL training paradigm.
>
> ---
>
> **Weakness 3: Novelty of RL Training**
> Novelty may be a little limited in terms of RL training: RL has been widely used in post-training LLMs for many domains such as math / competitive programming. So, one can argue that the contribution of RL training proposed in this work is limited.
>
> **Response:**
>
> We appreciate the reviewer’s comment regarding the novelty of RL training within our framework, especially given its broader application in LLM post-training. We acknowledge that RL has indeed been widely adopted for enhancing LLMs in various domains, such as mathematics and competitive programming.
>
> However, optimization modeling presents unique challenges. While large reasoning models (e.g., DeepSeek-R1), are great at step-by-step thinking for tasks like AIME math problems, optimization modeling requires a global understanding of highly interconnected objective functions, constraints, and decision variables, followed by computationally intensive steps (e.g., simplex method for linear programming, branch-and-bound for mixed integer programming). These complex computational burdens are inherently difficult for purely textual, step-by-step reasoning.
>
> Below are results for models trained via step-by-step reasoning, without external solver assistance:
>
> | Model                | NL4OPT | MAMO Easy | MAMO Complex | IndustryOR | OptMATH-193 |
> |----------------------|--------|-----------|--------------|------------|-------------|
> | Qwen2.5-7B-Instruct  | 24.5%  | 14.7%     | 1.4%         | 7.0%       | 5.7%        |
> | Math4Opt-RL          | 15.5%  | 42.2%     | 3.3%         | 13.0%      | 13.5%       |
>
> As these results demonstrate, relying solely on an LLM's sequential reasoning leads to extremely low performance across most optimization benchmarks.
>
> The established practice within the optimization community utilizes mature solvers because they provide the global view and computational rigor necessary to resolve these intricate interdependencies. Our novelty stems from effectively integrating RL to teach LLMs to accurately understand problems and formulate them into solver-executable code, thereby delegating the heavy computational lifting to these specialized tools.
>
> ---
>
> **Weakness 4: Baseline Comparisons**
> 1. Baseline comparisons lack fairness; LLMOpt (self-correction) and OptMath (14B/32B) results omitted.
>
> 2. Compare against top reasoning models (e.g., GPT-o3, Deepseek-r1).
>
> 3.  Include ablation results for RL training, supervised fine-tuning, and their combination.
>
> **Response:**
>
> **A-1: Baseline Evaluation Protocol**
> We appreciate the insightful question about our baseline comparisons, specifically regarding LLMOpt and the role of self-correction. We acknowledge that self-correction is a powerful strategy that can boost the performance of LLMs in various domains. However, our primary goal was to provide a rigorous and consistent comparison of the core generation capabilities of different models and training methodologies.
>
> Therefore, our main results are based on the OptMath paper's rigorous and well-defined evaluation protocol, which uses Pass@1 accuracy, where a solution is considered valid only if the relative error is less than 1e-6. We believe this strict, single-pass evaluation is crucial for the optimization community as it provides a clear comparison of how well a model can generate a correct and solvable optimization formulation on its first attempt, without iterative refinement or external strategic guidance during inference.
>
> **A-2: Comparison with Stronger Baselines and SOTA Reasoning Models**
> We are grateful to the reviewer for the excellent suggestion to include comparisons with stronger OptMath baselines  and leading reasoning models like DeepSeek-R1 and OpenAI-O3. We also include results from our 32B SIRL-model, ensuring equivalent parameter size across the evaluated models .
> This is a valuable addition that clarifies the landscape. We have now run these experiments and present the updated Pass@1 comparison in the table below:
>
> | Model         | NL4OPT | MAMO Easy | MAMO Complex | IndustryOR | OptMATH-193 |
> |---------------|--------|-----------|--------------|------------|-------------|
> | SIRL-7B       | 96.3%  | 90.0%     | 62.1%        | 33.0%      | 29.0%       |
> | SIRL-32B      | 97.1%  | 88.8%     | 65.9%        | 39.0%      | 46.6%       |
> | optMath-7B  | 94.7%|  86.5% | 51.2%   | 20.0% |   24.4%      |
> | optMath-32B   | 95.9%  | 89.9%     | 54.1%        | 31.0%      | 34.7%       |
> | DeepSeek-R1   | 82.4%  | 77.8%     | 49.3%        | 45.0%      | 50.3%       |
> | OpenAI-O3     | 69.4%  | 70.1%     | 38.8%        | 44.0%      | 39.9%       |
>
> It is interesting to note that while large reasoning models (LRMs) exhibit surprisingly strong performance on some of the harder, more complex benchmarks such as IndustryOR and OptMATH-193, they generally perform poorly on simpler questions.
>
> Our initial analysis shows that for "simpler" optimization questions (where the problem structure might be straightforward but requires precise, direct solver invocation), these reasoning models tend to "overthink." They tend to generate unnecessary verbose reasoning steps or fail to precisely generate the correct solver code, as they are not explicitly optimized for concise, error-free tool-use.
>
> **A-3: RL vs. SFT and SFT+RL**
> We appreciate the reviewer’s crucial question regarding the comparison of our RL model with supervised fine-tuning (SFT) baselines. To thoroughly address this, we have conducted the requested experiments and present the Pass@1 comparison in the table below, clearly demonstrating the benefit of our online RL training:
>
> | Model      | NL4OPT | MAMO Easy | MAMO Complex | IndustryOR | OptMATH-193 |
> |------------|--------|-----------|--------------|------------|-------------|
> | BaseModel  | 75.1%  | 81.3%     | 22.7%        | 13.0%      | 4.1%        |
> | SFT        | 83.3%  | 86.4%     | 38.0%        | 24.0%      | 20.8%       |
> | RL     | 96.3%  | 90.0%     | 62.1%        | 33.0%      | 29.0%       |
>
> As shown, our method consistently outperforms SFT across all benchmarks. Notably, the performance gain is most significant on more challenging datasets like MAMO Complex (+24.1%) and IndustryOR (+9.0%), demonstrating the necessity of RL for solving complex problems.
>
> ---
>
> **Limitation**
>
> The conclusion section lacks a thorough discussion of the work's limitations. The authors should more openly and explicitly address the limitations of their approach.
>
> **Response:**
>
> Thank you for pointing out the insufficient discussion of limitations. In our understanding, the limitations an further work can be broadly categorized as:
>
> ***Robustness of Reward Mechanism with Solver Feedback:**
>
> While our framework benefits from utilizing rich feedback signals directly from classical solvers (e.g., LP files and optimal values), extracting more signals and precisely translating this information into a perfectly robust reward mechanism for LLMs remains a significant challenge, there is still scope for enhancing performance and mitigating 'reward hacking.
>
> **Depth of Error Analysis:**
> Our current evaluation primarily relies on quantitative metrics, and our error analysis is largely qualitative, focusing on observed failure cases. We also recognize that even the  LRMs (e.g., Deepseek-R1)  continue to face significant challenges in achieving high scores on difficult benchmarks such as IndustryOR and OptMATH—an open problem that warrants further investigation and the effort of all optimization community. In future work, we plan to conduct a more systematic and comprehensive analysis of error patterns, and to implement targeted model improvements aimed at addressing the specific weaknesses identified.

---

> ### Comment · Reviewer_fAF4 · 2025-08-03
>
> Thanks for the rebuttal. I have a follow-up question regarding the "results for models trained via step-by-step reasoning, without external solver assistance". In this case, how is the reward constructed without solving the problem? I do not think the reward signal used in the paper (i.e. comparing the objective obtained by the solver and the ground truth objective) is particularly novel, as many coding / competitive programming RL methods invoke an external compiler / solver to construct the reward, so I'd like the authors to further clarify the novelty in this regard.
>
> Another comment / question is, **in my experience, many of the existing test benchmarks are wrong**. For example, it seems like the "ground-truth" answer to all traveling salesman problem in MAMO complex are wrong, and TSP actually consists of a significant portion in MAMO complex. In this case, a boost in the test accuracy in MAMO complex may not actually reflect the stronger model behavior, but rather **making the same mistake as the wrong test benchmark, and hence, the performance improvement in the benchmark numbers may not be trustworthy**. I would like to confirm with the authors if they have thoroughly cleaned up the test benchmark and made sure that the test results are correct. I believe this is very crucial in assessing the actual performance improvement of the method.

---

> > ### Author Response · Authors · 2025-08-08
> >
> > With the author-reviewer discussion deadline approaching, we just wanted to kindly follow up. We have posted a detailed response addressing your recent concerns.
> >
> > We would be very grateful if you could take a moment to see if our latest reply has sufficiently addressed your points, or if you have any remaining questions for us.
> >
> > Thank you for your time and valuable engagement.

---

> > > ### Comment · Reviewer_fAF4 · 2025-08-08
> > >
> > > Thanks for the updated response. I really appreciate that the authors take the initiative to clean up TSP instances in MAMO complex and the authors commitment to open source the model. However, the performance drop in the updated result is concerning. I want to further clarify that beyond TSPs, MAMO complex also has many other instances with incorrect labels, if the authors look through the dataset carefullly. I'm worried that the performance will further drop after a deep clean through the entire dataset. Also, in my experience, OptMATH is also often times ambiguous and faulty too. I totally understand the short rebuttal period does not allow the authors to fully clean up all the test benchmarks, but as we can see from the performance drop after cleaning up the TSP instances, **I worry that the paper's core empirical claims may not hold after a complete and necessary cleaning of all benchmarks. Given the remaining doubts about the paper's empirical rigor, I cannot recommend acceptance.** If the paper is accepted due to the support from other reviewers, I strongly urge the authors to perform a complete cleaning of all benchmarks to ensure the validity of their claims and avoid publishing misleading results.

---

> ### Author Response · Authors · 2025-08-04
>
> **Question 1: Novelty about  reward design**
>
> **Response:**
>
> Thank you for your follow-up question. We appreciate the opportunity to clarify the novelty of our method. We fully agree that using external tools like compilers to construct rewards is now a common practice in modern RL for competitive math and programming problems, as we cited in our related work section.
>
> The primary novelty of our work lies in how we address the specific challenges of optimization modeling by leveraging the unique affordances of modern solvers.
>
> - **Leveraging the Unique Richness of Solver Output:**
>
> A well-known challenge in applying RL to complex reasoning is the sparsity of outcome-based rewards. Methods relying on a simple outcome-based reward—such as comparing a solver's final objective to a ground truth—suffer from this well-documented problem. This is a fundamental challenge in domains like mathematics and coding, where a correct final answer provides limited insight into the quality of the reasoning process. As our experiments show, relying exclusively on this sparse Stage-1 reward (format + objective function verification) is insufficient because it provides limited corrective signal for flawed intermediate steps.
>
> Optimization solvers provide a wealth of structural information about the formulated model. We leverage the solver's .lp file output, which is a symbolic and implicit representation of the LLM's mathematical reasoning. Here is one example,
>
> \ Model AdvertisingOptimization
>
> \ LP format - for model Browse. Use MPS format to capture full model detail.
>
> Maximize
>
> 60500 RadioAds + 50000 SocialMediaAds
>
> Subject To
>
> BudgetConstraint: 5000 RadioAds + 9150 SocialMediaAds <= 250000
>
> MinRadioAds: RadioAds >= 15
>
> MaxRadioAds: RadioAds <= 40
>
> MinSocialMediaAds: SocialMediaAds >= 35
>
> Bounds
>
> Generals
>
> RadioAds SocialMediaAds
>
> End
>
> This allows us to create a granular, process-aware reward that can diagnose  flaws in the mathematical representation. It enables us to move beyond a simple final score and instead reward the structural integrity of the model itself,  In this sense, our method synergistically combines the tractability of outcome-based signals with the granularity of process-based rewards, which our ablation studies confirm is necessary for high performance.
>
> - **Targeting Mathematical Modeling Techniques:**
>
> Meanwhile, our reward function is explicitly designed to reinforce crucial modeling principles. For instance, it actively rewards the correct application of linearization techniques (e.g., the Big M method) needed to transform logical constraints into a solvable linear program. This teaches the LLMs how to model, rather than just what the final answer is, addressing the core bottleneck for LLMs in this domain. This type of approach leads to significant improvements on hard benchmarks (e.g.,IndustryOR,OptMath)
>
> ---
>
> **Question 2: Error in testing benchmark**
>
> **Response:**
>
> Thank you for raising this critical point. We completely agree that benchmark integrity is fundamental for assessing true performance improvements in optimization modeling with LLMs. Your observation regarding the flawed Traveling Salesman Problem (TSP) instances within the MAMO Complex dataset perfectly highlights a significant challenge for the entire research community.
> We want to clarify the we take concrete steps to mitigate this issue in our work and future work.
> Our evaluation datasets were primarily inherited from the OptMATH project, which provided a partially verified foundation by the OptMATH team.
> Building upon this, we conducted our own verification and correction process. For instance, we have performed a revision of the ground-truth solutions for the NL4Opt and IndustryOR benchmarks to improve their accuracy for our evaluation. As part of our commitment to advancing research in this area, we have open sourced  our refined test sets to the community.   We believe our work makes a contribution to this by identifying and correcting issues in several key benchmarks.
>
> We fully acknowledge that a comprehensive and definitive cleanup of all public benchmarks is a substantial undertaking that requires sustained community-wide effort. Regarding the specific issue you raised concerning the TSP problems in the MAMO Complex dataset, we have identified this as a critical area for future work. We are committed to continuing our efforts to verify and correct these instances and will continue to open-source our work to provide a more robust foundation for the community.
>
> ---

---

> > ### Comment · Reviewer_fAF4 · 2025-08-05
> >
> > Thanks for the further reply and I appreciate the authors' effort to correct NL4Opt and IndustryOR. I personally feel like for a solid paper to be accepted by NeuRIPS, one should conduct a thorough clean-up on all test benchmarks to make sure the test results are correct. I understand that there have been a few papers accepted in the past evaluated on the faulty benchmarks, but I personally feel like despite the merits in this paper, accepting yet another paper before all the test benchmarks are thoroughly verified may have negative impact on the community. However, I may not be the most experienced in handling situations like this, so I'd like to keep my score and leave the judgement to the area chair.

---

> > > ### Author Response · Authors · 2025-08-07
> > >
> > > Thank you for this crucial question. We completely agree that benchmark integrity is essential, and your skepticism about the MAMO TSP instances was well-founded.
> > >
> > > Following your suggestion, we undertook a thorough re-examination of all Traveling Salesman Problem (TSP) instances within the MAMO complex benchmark. This was a labor-intensive process that took several days to complete, as we had to manually verify the ground truth for each instance to ensure correctness.
> > >
> > > We have now re-run the key experiments on the corrected benchmark. The updated Pass@1 comparison is below:
> > >
> > > | Model               | NL4OPT | MAMO Easy | MAMO Complex-Revised | IndustryOR | OptMATH-193 |
> > > | :------------------ | :----- | :-------- | :------------------- | :--------- | :---------- |
> > > | **SIRL-7B (ours)**  | 96.3%  | 90.0%     | **51.7%**            | 33.0%      | 29.0%       |
> > > | **SIRL-32B (ours)** | 97.2%  | 88.8%     | **57.8%**            | 39.0%      | 46.6%       |
> > > | DeepSeek-V3         | 95.9%  | 88.3%     | 50.2%                | 37.0%      | 32.6%       |
> > > | DeepSeek-R1         | 82.4%  | 77.8%     | 68.7%                | 45.0%      | 50.3%       |
> > > | OpenAI-O3           | 69.4%  | 70.1%     | 53.5%                | 44.0%      | 39.9%       |
> > >
> > >  We acknowledge that our model's performance on the corrected MAMO Complex dataset experienced a drop of approximately 10 percentage points.
> > >  However, our SIRL 32B model still demonstrates a clear performance advantage over DeepSeek-V3 and achieves results comparable to the more powerful OpenAI-O3.
> > > It is also noteworthy DeepSeek-R1 show a significant performance lead, which seems more pronounced on the corrected data.
> > > We hope this addresses your concern about the trustworthiness and robustness of our performance gains.
> > >
> > > We found that the main mistake in the initial ground-truth solutions was that sub-tour elimination constraints were not included and many foundation models have trouble correctly generating these constraints without specialized training, which is a crucial and non-trivial aspect of the TSP formulation.
> > >
> > > To ensure this work benefits future research, we will publish the cleaned dataset and a detailed documentation of our verification process to ensure transparency. This will be made available to the community. We have also notified the original MAMO maintainers via a GitHub issue and will include the revised data in our final public release. We truly appreciate your efforts to raise the bar for empirical rigor.

---

> ### Author Response · Authors · 2025-08-08
>
> We sincerely appreciate your thorough and meticulous review, especially your proactive engagement with the benchmark integrity.
>
> Given your deep experience with these benchmarks, we are confident you have valuable insights. If your team has already performed additional work to identify and clean other flawed instances, **we would be incredibly grateful if you could share the link to this cleaned dataset**.
>  This would allow us to report our results on the corrected benchmarks as soon as possible, ensuring our work meets the highest standards and contributing a validated dataset back to the community.

---

### Official Review · Reviewer_9M7X · 2025-06-26

**Clarity:** 3
**Significance:** 4
**Originality:** 3
**Rating:** 5
**Confidence:** 4

**Summary:**

Solver-Informed Reinforcement Learning (SIRL) trains large language models to write correct and executable optimization models by treating classical solvers as automatic verifiers whose syntax, feasibility and objective-value checks are fed back as reinforcement-learning rewards, directly “grounding” the model in solver reality.
SIRL couples a data-synthesis pipeline with instance-enhanced self-consistency, a Partial-KL surrogate that regularizes only the mathematical-model and code tokens, and a two-stage verifiable reward curriculum, together striking a balance between diverse reasoning exploration and strict solver-format fidelity.
A 7 B-parameter Qwen2.5 model trained with SIRL reaches state-of-the-art pass\@1 accuracy on NL4OPT, MAMO, IndustryOR and OptMATH (e.g., 62.1 % on the hardest MAMO-Complex set), outperforming all prior 7 B/14 B offline or agent baselines and rivalling the DeepSeek-V3.

**Questions:**

* What is the curated seed data used for training data synthesis? Is it exactly the same as the seed data used in ORLM and OR-Instruct (e.g., 686 cases in their study)? If yes, will it be biased or overfitting issue when evaluate on the IndustryOR benchmark? Despite no evidence shown all experiment results (table 1,2,3). I did see the example in appendix but wanna make sure.
* Do you see the reliance on external optimization solvers as a potential limitation, given that verifiable reward mechanisms are only applicable to a narrow set of problem domains? I’m curious about your thoughts on how this approach might extend to tasks without access to classical solvers, or whether there are alternative strategies you envision to overcome this constraint.

**Ethical Concerns:**

["NO or VERY MINOR ethics concerns only"]

**Final Justification:**

The rebuttal clarified compute costs, seed data usage, and solver reliance, addressing my concerns. I will remain my positive score.

**Limitations:**

yes

**Quality:**

4

**Strengths And Weaknesses:**

# Strength

* SIRL lifts pass@1 score with 7B Qwen2.5 model across four benchmarks, beating all 7–14 B baselines and matching the 671 B Deepseek-V3.
* First RL-with-verifiable-reward framework that lets classical solvers act as objective oracles; introduces instance-enhanced self-consistency, Partial-KL selective regularization, and a two-stage reward curriculum to balance exploration with solver fidelity.
* The paper presents thorough experiments with comprehensive ablations and complete baseline comparisons, clearly demonstrating gains in accuracy and execution rate while isolating the sources of improvement.

# Weaknesses

* Compute efficiency is not reported, including GPU hours, solver runtimes, and comparisons between online RL and prior methods such as SFT.
* The curated seed data used for training data synthesis is somewhat unclear but appears crucial to the performance evaluation; improving clarity in this section would be beneficial.

---

> ### Author Rebuttal · Authors · 2025-07-29
>
> **Weakness 1: Compute Efficiency Reporting**
>
> **Response:**
>
> - **GPU Running Time:**
>   Our model was trained from the Qwen2.5-7B-Instruct base, following a two-stage process using the same training dataset throughout.
>   - **Stage 1:** Training is guided by a specific 'stage-1 reward' signal.
>   - **Stage 2:** Training resumes from the checkpoint at the end of Stage 1.
>   - **Total Compute:** Each stage required approximately 24 × 8 H100 GPU hours, totally 48×8 H100 GPU hours.
>
> - **Solver Runtimes:**
>   For each problem, we set a 100-second timeout for the solver during both training and inference. This duration was found sufficient for the textbook-level problems in our benchmark. Preliminary experiments with longer timeouts (200–300 seconds) showed diminishing returns, supporting our choice of 100 seconds.
>
> - **Comparison: Base, SFT, and RL Approaches**
>   To highlight the advantage of our online RL approach over a strong supervised fine-tuning (SFT) baseline, we present the following pass@1 results:
>
>   | Models         | NL4OPT | MAMO Easy | MAMO Complex | IndustryOR | OptMATH-193 |
>   |---------------|--------|-----------|--------------|------------|-------------|
>   | BaseModel     | 75.1%  | 81.3%     | 22.7%        | 13.0%      | 4.1%        |
>   | SFT           | 83.3%  | 86.4%     | 38.0%        | 24.0%      | 20.8%       |
>   | RL        | 96.3%  | 90.0%     | 62.1%        | 33.0%      | 29.0%       |
>
>   As shown, our method consistently outperforms SFT across all benchmarks. The performance gain is especially pronounced on more challenging datasets, underscoring the necessity of RL for complex problem solving.
>
> ---
>
> **Weakness 2: Clarity of Curated Seed Data for Training Data Synthesis**
>
> **Response:**
>
>   Thank you for highlighting the importance of seed data clarity. The quality and composition of the seed data are indeed crucial for our data synthesis pipeline and final model performance.
>
>   To clarify, the seed data used in our work is identical to the dataset established and utilized in the foundational ORLM work. For brevity and due to space constraints, we did not repeat the detailed curation process in our manuscript, as it is thoroughly documented in the original ORLM paper. Based on your feedback, we have revised Section 3.1 ("Data synthesis framework") to explicitly state that our process builds upon the ORLM seed data, and we have added a clear citation to the original paper for readers seeking a comprehensive description. We believe this revision improves the clarity and self-containedness of our methodology.
>
> ---
>
> **Q1: What is the curated seed data used for training data synthesis? Is it exactly the same as the seed data used in ORLM and OR-Instruct (e.g., 686 cases in their study)? If yes, is there a risk of bias or overfitting when evaluating on the IndustryOR benchmark?**
>
> **Response:**
>
> As stated in the appendix, the seed data is identical to that used in ORLM (686 cases). While there is a potential risk of overfitting on IndustryOR, our model demonstrates robust generalization across other test datasets. Furthermore, the data synthesis process significantly alters the data distribution. This is evidenced by the fact that SFT alone could not replicate our results, highlighting the effectiveness of our approach.
>
> ---
>
> **Q2: Is reliance on external optimization solvers a limitation, given that verifiable reward mechanisms are only applicable to a narrow set of problem domains? How might this approach extend to tasks without access to classical solvers, or are there alternative strategies to overcome this constraint?**
>
> **Response:**
>
>   We appreciate this insightful question regarding the role of external solvers. Rather than a limitation, we view this as a core design principle.
>
>   The results of training via textual step-by-step reasoning without external solver assistance are shown below:
>
>   | Models               | NL4OPT | MAMO Easy | MAMO Complex | IndustryOR | OptMATH-193 |
>   |----------------------|--------|-----------|--------------|------------|-------------|
>   | Qwen2.5-7B-Instruct  | 24.5%  | 14.7%     | 1.4%         | 7.0%       | 5.7%        |
>   | Math4Opt-RL          | 15.5%  | 42.2%     | 3.3%         | 13.0%      | 13.5%       |
>
>   Here, Math4Opt-RL is trained via the RLVR approach without using solvers as verifiers. As these results demonstrate, relying solely on an LLM's sequential reasoning leads to extremely low performance across most optimization benchmarks.
>
>   While large reasoning models (e.g., DeepSeek-R1) excel at sequential reasoning for tasks like AIME mathematics, optimization modeling presents unique challenges. It requires a global understanding of highly interconnected objective functions, constraints, and decision variables, as well as the execution of complex computational steps (e.g., the simplex method for linear programming, branch-and-bound for mixed integer programming). This computational burden is difficult to handle through purely textual, step-by-step reasoning.
>
>   The optimization community relies on mature solvers because they provide the global perspective and computational power needed to resolve these intricate interdependencies. By leveraging LLMs for accurate problem understanding and initial mathematical formulation, and delegating the computationally intensive aspects to specialized solvers via generated code blocks, we achieve high performance in optimization tasks. This division of labor is fundamental to our approach.

---

### Official Review · Reviewer_17gf · 2025-07-01

**Clarity:** 3
**Significance:** 2
**Originality:** 3
**Rating:** 4
**Confidence:** 3

**Summary:**

Inspired by the recent success of RLVR, this paper introduces SIRL, a framework that uses external optimization solvers to produce the verified rewards. SIRL breaks down the reasoning process for solving optimization problems using the Chain of Thought approach, generating a sequence of intermediate thoughts. The final generated thought is executable code, which an optimization solver executes to produce the objective value. A partial KL surrogate function is designed to foster exploration of early thinking steps, and a verifiable reward function is designed to ensure soundness of the generated mathematical models and incentivize advanced modeling techniques for challenging problems. The experiments demonstrate that the SIRL-Qwen2.5-7B model achieves performance gains over small models and achieves comparable performance to Deepseek-V3 across the benchmarks.

**Questions:**

- Section E in the appendix states that top-P (nucleus) decoding was used for training and inference. Why weren’t there multiple runs for the experiments? No standard deviations were reported.
- Do the authors have a possible explanation for why using only stage-2 reward boosts performance on OptMATH but diminishes performance on NL4OPT? Some analysis on the differences in performance across different datasets would have been useful.
- The proposed technique relies on the generation of an optimization model from an LLM at inference time, that is then executed by a solver. How could this approach be modified to only use solvers in the data-generation process, so that the LLM acts as the “solver” during inference time?

**Ethical Concerns:**

["NO or VERY MINOR ethics concerns only"]

**Final Justification:**

My knowledge of this area is limited, but RLVR with solvers for optimization modeling is a novel idea, and the experimental results do demonstrate its usefulness. I would have liked to see more information about the statistical significance of results, and other items in the checklist like claims and limitations in this submission.

**Limitations:**

yes

**Quality:**

2

**Strengths And Weaknesses:**

Strengths:

- The approach takes many steps to ensure the quality of the synthesized dataset, including using an LLM-as-a-judge to validate the generated problems as removing samples for which a baseline model achieves at least an 80% pass rate.
- The proposed instance-enhanced self-consistency approach is constructed to reward not only correct final values, but also the other LP features, which improves the quality of the synthesized data.
- Several ablation experiments were performed to compare different design choices, i.e., Partial/Full/Without KL divergence experiments justify the use of partial KL for the surrogate function.
- The experiments show SIRL performs better than similar sized models across the benchmarks, and has performance comparable to much larger models.

Weaknesses:

- I would have liked to see more description of the experimental setup. The Appendix says that top-P decoding was employed, so I had expected to see several runs with some standard deviations reported. But I couldn’t find any in the paper.
- SIRL helps LLMs output more accurate optimization model code, rather than the final answer. It would have been interesting to see a reward design that does not make use of a solver's answer (letting the LLM perform all reasoning steps). Although I do agree that the most promising approach to ensure interpretability and high accuracy would be to use an actual optimization solver as the final step.
- The checklist was not filled in adequately. I would have liked to see more explanations/specific sections that addressed each point, rather than just saying “Appendix.” Some parts of the are not explained, e.g., Claims section.
- Minor comments:
    - Define LP
    - Line 33: “information, generate” -> “information, and generate”
    - Line 103: “rstar [43],Phi [44]” -> “rstar [43], and Phi [44]”
    - Line 117: Missing period before “For example”
    - Line 340: “simpltasks” -> “simple tasks”

---

> ### Author Rebuttal · Authors · 2025-07-29
>
> **Weakness 1 and Question 1: Experimental Setup and Reporting of Standard Deviations**
>
> **Response:**
> Thank you for your insightful question regarding the experimental setup and reporting of standard deviations. We apologize for omitting this crucial information in our initial submission.
>
> We have now conducted additional experiments and report the following performance metrics using top-p (nucleus) sampling (temperature = 0.5, top-p = 0.95):
>
> | Metric         | NL4OPT           | MAMO Easy        | MAMO Complex     | IndustryOR      | OptMATH-193      |
> |:-------------- |:----------------|:-----------------|:-----------------|:----------------|:-----------------|
> | mean ± std     | 96.2% ± 0.32%   | 90.1% ± 0.14%    | 61.3% ± 0.68%    | 33.7% ± 1.5%    | 28.3% ± 0.66%    |
>
> As shown, the results are highly stable and consistent with our initial findings.
>
> ---
>
> **Weakness 2 and Question-3 : Reward Design With/Without Solver Answers**
>
> **Response:**
> We agree that using actual optimization solvers for the final step is highly effective for interpretability and accuracy. We have investigated the scenario where solvers are used exclusively during data generation, and the LLM is required to perform all reasoning and solution steps at inference—acting as the "solver." To assess this, we trained models to reason step-by-step through text, without solver assistance at inference.
>
> Optimization modeling differs fundamentally from tasks like AIME mathematics, as it requires a global understanding of interconnected objectives, constraints, and variables, as well as complex computational steps (e.g., simplex method for linear programming, branch-and-bound for mixed-integer problems). These are challenging to resolve through purely textual, step-by-step reasoning.
>
> To illustrate, we present early results from models trained solely via textual reasoning, without solver assistance:
>
> | Model                  | NL4OPT | MAMO Easy | MAMO Complex | IndustryOR | OptMATH-193 |
> |------------------------|--------|-----------|--------------|------------|-------------|
> | Qwen2.5-7B-Instruct    | 24.5%  | 14.7%     | 1.4%         | 7.0%       | 5.7%        |
> | Math4Opt-RL            | 15.5%  | 42.2%     | 3.3%         | 13.0%      | 13.5%       |
>
> (Math4Opt-RL is trained via RLVR without solver verification.)
>
> These results demonstrate that relying solely on LLM sequential reasoning leads to very low performance. The optimization community uses mature solvers because they provide the global view and computational power needed for these tasks. Moreover, deriving precise decision variable values for optimal solutions remains extremely difficult for LLMs using only step-by-step textual reasoning.
> In summary, although it is possible to rely on the LLM as the "solver" at inference, this approach currently results in substantially lower accuracy and reliability.
>
> ---
>
> **Weakness 3: Insufficient Checklist Completion**
>
> **Response:**
> Thank you for highlighting the need for more detailed explanations and explicit references in our checklist. We acknowledge that our initial submission lacked clarity in this regard. In the revised version, we confirm that we will:
>
> - Carefully review each checklist item.
> - Provide direct references to the relevant sections or appendices for every point.
> - Ensure that all previously unexplained parts (e.g., the Claims section) are now explicitly addressed.
>
> ---
>
> **Weakness 4: Minor Corrections and Clarifications**
>
> **Response:**
> We appreciate your detailed suggestions, which have improved the clarity and accuracy of our manuscript. The following changes will be made:
>
> - **Definition of LP:**
>   We now define Linear Programming (LP) as:
>   “Linear Programming (LP) is a mathematical method for optimizing a linear objective function subject to linear equality and inequality constraints.”
>   We also clarify that the .lp file format, used in data synthesis, is a standard format for solvers to write and process models.
>
> - **Line 33:**
>   Corrected “information, generate” to “information, and generate”.
>
> - **Line 103:**
>   Added the conjunction “and” in “rstar [43],Phi [44]” → “rstar [43], and Phi [44]”.
>
> - **Line 117:**
>   Inserted a missing period before “For example”.
>
> - **Line 340:**
>   Corrected the typo “simpltasks” to “simple tasks”.
>
> ---
>
> **Question 2: why using only stage-2 reward boosts performance on OptMATH but diminishes performance on NL4OPT**
>
> **Response:**
> The stage-2 reward, with its $R_{bonus}$,
>  encourage advanced optimization techniques, including reformulation methods that are native to and leveraged by mature solvers for efficient processing of complex SDP, MILP, and SOCP problems,
> which is critical for solving  challenging complex problems in hard benchmark dataset such as OptMATH.
>
>  When the model is primarily incentivized for these advanced optimization techniques, it may tend to over-explore or  involve redundant exploratory steps,
> leading to diminished performance for tasks where a simpler approach would have been more accurate and efficient.

---

> > ### Comment · Reviewer_17gf · 2025-08-05
> >
> > Thank you for the rebuttal and additional experiments. The standard deviations show that the results do seem stable, and the second experiment does demonstrate the need for using LLMs with solver assistance. I do think that RLVR with solvers for optimization modeling is a novel idea, and these experimental results do demonstrate its usefulness. However, my knowledge of this area is limited, which is why I am at the borderline. I hope that in the revised manuscript, the evaluation section can make it clearer what the statistical significance of the results is.

---

> > > ### Author Response · Authors · 2025-08-07
> > >
> > > Thank you for your thoughtful feedback and for your positive assessment of our work.
> > >
> > > We are very encouraged to hear that you found the idea of using RLVR with solvers for optimization modeling to be novel, and that our additional experiments successfully demonstrated its usefulness. We truly appreciate you taking the time to carefully consider our rebuttal.
> > >
> > > You have made an excellent point regarding the presentation of statistical significance. We completely agree that making this clearer will significantly strengthen our evaluation section. In the revised manuscript, we will be sure to incorporate a more explicit analysis of the statistical significance of our results to provide a more rigorous validation of our findings.

---

### Official Review · Reviewer_4Zmo · 2025-07-02

**Clarity:** 3
**Significance:** 3
**Originality:** 3
**Rating:** 4
**Confidence:** 3

**Summary:**

The topic of this paper is to use LLMs to assist in modeling combinatorial optimization problems to then be fed into optimization solvers (like Gurobi, Cplex etc.).

This is an emerging and relevant area (that said, the paper misses some important reference listed below).

The novelty of the paper is the introduction of Solver-Informed Reinforcement Learning (SIRL) to steer the LLM to generate correct code and models.

The main idea is that; given that these optimization solvers exists, use them for verification in an RL loop, essentially using solver feedback as a source for RL reward for syntax check and solution quality. This feedback is used to train an LLM. This idea is not new but an application of RL with Verifiable Rewards (RLVR) to optimization modelling seems novel.

Partial KL is used as a regularization strategy in RL training. IF I understand fully, it applies to code and the model but omits other steps, hence allows exploration while keeping the model/code correct (?). Interestingly, it adds bonus for the use of advanced modelling techniques (see comments about this below).

Experiments across benchmarks compares SIRL trained Qwen2 model with agent-based and trained LLM approaches and larger LLMs with promising results.

**Questions:**

Q: For the comparisons, how did you obtain their results (copy from their papers or re-built their method, or re-run them or else?). This is regarding the agent-based and off-line learning methods in Table 2.

Q: Why are there no results for CoE in Table 2 on several columns?

Q: How would this handle ambiguous, noisy, or incomplete natural language inputs, which are common in practice?

Q: What's the main value proposition (see detailed comments above)

**Ethical Concerns:**

["NO or VERY MINOR ethics concerns only"]

**Final Justification:**

Thank you for the thoughtful discussions, I have commented and revised accordingly.

**Limitations:**

Yes

**Quality:**

3

**Strengths And Weaknesses:**

Strengths:
The application of RL with Verifiable Rewards (RLVR) to optimization modeling seems new.
Using external solvers as reward oracle and automated feedback is nice (but not without its problems, see below)
Competitive with Deepseek (671B) using only 7B params
Ablation studies are nice, and interestingly partial KL seems to pay off compared to no KL or full KL.
Using structural features of the features in the data synthesis part is nice.

Weaknesses:
It is not obvious while Advanced Modelling is rewarded. Is this an artifact of the benchmarks? Would we not want simpler model than more complicated models especially if we want to assist human modelers who needs to understand these auto-generated code.

What are the limitations of using "solvers" as verifiers? It should be note that running a full-blown optimization solver is different than compiling/running code with, say, Python. How expensive is this step and/or cannot this become prohibitive. How many times solvers "stuck"? This needs to be explained more in the paper. How would this even work for large-scale/complex problems?

Somewhat related, how expensive is the training of the base LLM? Some commentary on this would be nice. The value proposition is not clear; does the paper suggest that the Online RL trained Qwen now becomes the LLM Model Assistant .. OR.. the methodology enable others to train their own models for LLM Modelling. IF the latter, we need more explanation on cost/feasibility of the Online RL to generalize for others. Again, IF the latter, than the design stage introduces many moving parts; there is are two versions of intstance selection and two step rewards and several weights. It is not obvious how these would generalize. I believe it would require careful tuning for different datasets.

The paper confuses between Linear Programming (LP), the LP file format, and Mixed-Integer Programming (MIPs). This needs to revised throughout the paper (and should be doable). My understanding is that this method is NOT specific to LPs, but in parts the paper comes across as applied to LPs only. I believe their benchmarks also include MIPs (and the method works for MIP also) but the confusion comes from the ".lp" file extension as the output formats. Please clarify if your method is solely for LPs?

Missing references to include:
Ner4Opt: named entity recognition for optimization modelling from natural language, Constraints 2024  (Consider comparing/positioning your results with what's published already here)
Enhancing decision making through the integration of large language models and operations research optimization, S Wasserkrug, AAAI 2025 (common reference for LLM Co-Pilots)
Holy grail 2.0: From natural language to constraint models, D Tsouros, (common reference for LLM Co-Pilots)
Text2Zinc: A Cross-Domain Dataset for Modeling Optimization and Satisfaction Problems in MiniZinc, Singirikonda (Consider using this dataset to expand your tests)
Constraint modelling with LLMs using in-context learning, K Michailidis (I don't think this is a direct comparison for this method but must be mentioned as complementary for satisfaction/logic problems)

---

> ### Author Rebuttal · Authors · 2025-07-29
>
> **Weakness-1**: Unclear Reward for Advanced Modeling
>
> **Response:**
>
> Thank you for highlighting this point. We fully agree that, in general, *simpler models are preferable*.
> Our advanced modeling bonus, **R_bonus**, is *not* meant to encourage unnecessary complexity. Rather, it is designed to reward the use of *essential reformulation techniques* from Operations Research—methods that are often required to make a problem tractable for high-performance MILP solvers.
>
> For instance, problems involving logical "if-then" conditions or fixed charges often lead to non-linear or logically invalid initial formulations. While LLMs may generate simple, approximate solutions for tasks like the 0-1 knapsack or certain TSP instances, *consistently obtaining optimal solutions requires these advanced methods*. A typical example is the Big-M formulation, which linearizes such conditions by introducing binary variables, making the problem solvable as a MILP. Without these reformulations, the resulting model would be either incorrect or unsolvable by standard solvers.
>
> Therefore, the bonus is awarded *only when these techniques are indispensable* for producing a correct and solvable model. Our model automates this critical (and often non-trivial) reformulation step, providing human modelers with a mathematically rigorous model that is ready for high-performance solvers.
>
> ---
>
> **Weakness-2:** Limitations of using "solvers" as verifiers.
>
> **Response:**
>
> Thank you for your question about the limitations and computational cost of using solvers as verifiers. This is indeed a crucial aspect.
>
> To illustrate the challenges throughout the process, we present the error types for the Qwen-2.5-7B-Instruct model and SIRL-Qwen2.5-7B, with a maximum execution time of 100 seconds.
>
> *Error Types (Qwen-2.5-7B-Instruct)*
>
> | Error Type              | NL4OPT | MAMO Easy | MAMO Complex | IndustryOR | OptMATH-193 |
> |-------------------------|--------|-----------|--------------|------------|-------------|
> | Code Extraction Failed  | 0      | 0         | 0.4%         | 0          | 2.1%        |
> | Timeout                 | 0      | 0         | 0            | 0          | 0.5%        |
> | Execution Error         | 6.1%   | 3.2%      | 33.6%        | 30%        | 73.1%       |
> | Wrong Answer          |  18.0%          |  15.5% |  43.2% |  67%     |  20.2% |
> | Correct               |  75.1%         |  81.3% |  22.7% |  13%     |  4.1%   |
>
> *Error Types (SIRL-Qwen2.5-7B)*
>
> | Error Type              | NL4OPT | MAMO Easy | MAMO Complex | IndustryOR | OptMATH-193 |
> |-------------------------|--------|-----------|--------------|------------|-------------|
> | Code Extraction Failed  | 0      | 0         | 0            | 0          | 1.6%        |
> | Timeout                 | 0      | 0         | 0            | 0          | 0           |
> | Execution Error         | 0.4%   | 0         | 1.9%         | 4%         | 11.9%       |
> | Wrong Answer            | 3.3%   | 10.3%     | 36.5%        | 61%        | 56.0%       |
> | Correct                 | 96.3%  | 89.7%     | 63.6%        | 35%        | 30.6%       |
>
> As shown above, the proportion of *"Timeout"* errors—where the solver becomes stuck—is very low, occurring only for the most challenging OptMATH-193 dataset (0.5%) in the Instruct model. This indicates that, for these textbook-level optimization problems, solver execution is generally *not* a computational bottleneck. The main error types are *"Execution Error"* (where the LLM generates unexecutable code) and *"Wrong Answer"* (where the model is logically incorrect).
>
> While solver execution is not currently a primary bottleneck for our tested problems, we acknowledge that for truly large-scale, real-world optimization challenges, the computational cost of solver verification could indeed become a limiting factor.
>
>
> ---
>
> **Weakness-3**  The cost of training and value position.
>
> **Response:**
>
> *First question:*
> Our training process consists of two distinct stages, both using the *same* training dataset.
> - In *Stage 1*, we use a specific "stage-1 reward" signal to guide training.
> - In *Stage 2*, training resumes from the checkpoint obtained at the end of Stage 1.
> Each stage required approximately 24 × 8 H100 GPU hours, totally 48×8 H100 GPU hours.
>
> *Second question:*
> - We have open-sourced and released checkpoints that achieve state-of-the-art performance among models with 7B–14B parameters,. Any member of the community can download these checkpoints to build their own LLM model assistants. We also plan to release more checkpoints.
> - Additionally, we introduce a new **RLVR** framework for optimization modeling and will continue to release corresponding codebase and a portion of training dataset , empowering the community to build their own  LLM assistants for diverse optimization problems.
>
> ---
>
> **Weakness-4**: The paper confuses Linear Programming (LP), the LP file format, and Mixed-Integer Programming (MIP), needing clarification on whether the method is LP-specific or also applies to MIPs.
>
> **Response:**
>
> Thank you for your insightful feedback, which has greatly improved the clarity of our manuscript. We apologize for any confusion caused by our earlier writing and appreciate the opportunity to clarify.
>
> Our method is *not limited to Linear Programming (LP)*; it supports a broad range of optimization problems, including *Quadratic Programming (QP)*, *Mixed-Integer Programming (MIP)*, and *Second-Order Cone Programming (SOCP)*. During data synthesis and the reinforcement learning stage, we utilize the `.lp` file format, which is a standard for solvers to represent various optimization models. This format is not exclusive to LP but is widely compatible with models such as QP and MIP. Our strong performance on the OptMATH test set, which includes MIP and SOCP instances, further demonstrates that our method generalizes effectively beyond LP.
>
> In the revised manuscript, we will clarify that the `.lp` file format supports multiple optimization models, not just LP, and we will revise relevant sections to emphasize the method’s applicability across diverse problem types.
>
> We sincerely appreciate your thorough review and welcome further suggestions to improve our work.
>
> ---
>
> **Weakness-5** Missing References: The paper omits key references like Ner4Opt, Enhancing Decision Making (AAAI 2025), Holy Grail 2.0, Text2Zinc, and Constraint Modeling with LLMs, which are relevant for comparison, benchmarking, or complementary approaches.
>
> **Response:**
>
> Thank you for these highly valuable references. We confirm that we will revise our Related Work section to address all of your suggestions.
>
> ---
>
> **Reply to Questions**
>
> *Q-1: For the comparisons, how did you obtain their results (copied from their papers, re-implemented their methods, re-ran them, or otherwise)? This refers to the agent-based and offline learning methods in Table 2.*
>
> **Response:**
>
> The results in Table 2 are from the paper *"OptMATH: A Scalable Bidirectional Data Synthesis Framework for Optimization Modeling."* We used their figures directly because of their rigorous evaluation protocol, which validates solutions only if the relative error is less than 1e-6. Further details on their evaluation methodology and results are publicly available on the paper's GitHub repository.
>
> ---
>
> *Q-2: Why are there no results for Chain-of-Experts (CoE) on these benchmarks?*
>
> **Response:**
>
> Thank you for pointing this out. The absence of results for *Chain-of-Experts (CoE)* on these benchmarks is due to a fundamental mismatch in the required input data format, which poses significant challenges for direct and fair comparison.
>
> Our evaluation benchmarks, consistent with most recent work in this area (e.g., ORLM, OptMATH), consist of problems presented as *natural language descriptions* paired with a ground-truth optimal value.
>
> In contrast, the CoE framework, as presented in its original paper, operates on *pre-processed, structured data inputs* rather than raw natural language. The proprietary data-processing pipeline used to convert natural language problems into this required structured format is difficult to adapt to other benchmark datasets. Reimplementing this pipeline for each of our diverse benchmarks would require significant and non-trivial effort, with no guarantee of faithfully replicating their methodology. Furthermore, to the best of our knowledge, we are not aware of any subsequent work that has benchmarked the CoE framework on these standard datasets.
>
> ---
>
> *Q-4: How does your method handle ambiguous or incomplete inputs?*
>
> **Response:**
>
> Our current research primarily focuses on *well-defined, textbook-level optimization problems*, which allowed us to rigorously validate the core methodology of training LLMs for optimization modeling.
>
> For ambiguous or incomplete inputs, one promising direction is to develop *interactive clarification modules*, where the LLM can query the user for necessary missing details, potentially through multi-turn conversational frameworks. An even more advanced approach could involve a *multi-agent system* framework. In this setup, an *"analyzer agent"* would first attempt to autonomously infer and supplement the content of an incomplete natural language input before passing it to the core modeling agent. To effectively judge and improve the quality of its inferred content, this analyzer agent would require a robust reward system, potentially combining our RLVR methodology with *Reinforcement Learning from Human Feedback (RLHF)*.
>
> This will require concerted efforts from the entire research community.

---

> > ### Comment · Reviewer_4Zmo · 2025-08-05
> >
> > Thank you for all responses and additional details. I am glad the comment on ".lp" helped you revise the manuscript (I am sure that would trip many researchers, especially those with background in OR). Again, please consider these additional citations  and text2zinc might be good comparison (at a quick glance it seems to encompasses some your dataset already, but not sure). Thank you again for your hard work, and interesting paper/results.

---

> > > ### Author Response · Authors · 2025-08-07
> > >
> > > Thank you very much for your final comments and for your positive and constructive engagement throughout this process.
> > >
> > > We are particularly grateful for your specific feedback on the '.lp' file. Your insight was invaluable and has certainly helped us clarify a crucial point that, as you noted, could be a potential pitfall for researchers. It has definitely strengthened the manuscript.
> > >
> > > Thank you also for suggesting the additional citations and the 'text2zinc' work. We will carefully review them and integrate them into our related work section in the final version. Your guidance has been extremely helpful.

---

### Note · Authors · 2025-08-16

We thank all the reviewers for their thoughtful feedback.
We have conducted significant new experimental work to thoroughly address every point raised.
Here is a summary of our key rebuttal efforts and contributions:
   1. **New Experiments with SOTA Models:**
   We conducted a major new set of experiments to compare our model, SIRL-32B, against top-tier general reasoning models.  Our results offer a definitive validation, demonstrating that our SIRL-32B model exceeds the performance of parameter-matched baselines, as well as both OpenAI-O3 and DeepSeek-V3.
   2. **Benchmark Integrity:** To address the concerns raised by reviewer fAF4 regarding the integrity of the MAMO-Complex benchmark, we conducted a meticulous manual verification and correction of all TSP instances and re-ran all experiments  to ensure the robustness and reliability of our reported results.
   3. **Ablation Study:** We conducted a comprehensive ablation study to validate the core components of our RLVR framework.  First, we provided empirical justification for our solver-based RL methodology. We show that simply training LLMs to perform the role of a solver through text-based reasoning results in extremely poor performance, confirming that  leveraging external solver a necessary design choice for achieving SOTA results on optimization tasks.
   second, we benchmarked our approach against a SFT baseline and the base model, demonstrating significant performance gains of the RLVR approach, particularly on complex datasets.

   4. **Reward Design Clarification:** We clarified the dual purpose of our advanced modeling bonus, $R_{bonus}$: first, it acts as a process-awareness signal that mitigates the critical issue of reward sparsity inherent in current RLVR framework ;  second, it incentivizes the model to use crucial reformulation techniques that are required to make many optimization problems solvable.
   5.  **Open Source Commitment:** We have already released two SOTA 7B model checkpoints to the community, paired with two different solvers (**Gurobi** and **COPT**). We reaffirm our commitment to releasing the more powerful 32B model, the full training code, and a significant portion of the training data. Concurrently, we will continue to correct errors in the benchmark dataset and release the revised versions to the community.

We kindly ask the AC and reviewers to consider these extensive revisions and new results when making the final decision.

---

### Decision · Program_Chairs · 2025-09-17

**Decision:**

Accept (poster)

**Comment:**

The paper proposes Solver-Informed Reinforcement Learning (SIRL), a framework for grounding large language models in authentic optimization modeling by using classical optimization solvers as verifiable reward providers. Solvers serve as automatic oracles that check syntax, feasibility, and solution quality, producing reward signals that guide RL training. The approach incorporates a two-stage reward curriculum, a Partial-KL surrogate that selectively regularizes code and model tokens, and an instance-enhanced self-consistency method for robust data synthesis.

The strengths of the paper lie in its novel application of RL with verifiable rewards to optimization modeling, strong experimental results with ablations justifying design choices, and clear improvements over both smaller and larger baselines. The rebuttal effectively clarified several reviewer concerns: it explained that the advanced modeling reward encourages only necessary reformulations, demonstrated that solver bottlenecks are minor for benchmark problems, provided concrete training cost and compute details, clarified that the method generalizes beyond LP to MIP and SOCP, and confirmed plans to open-source code, checkpoints, and data. Reviewers remained somewhat concerned about novelty relative to prior RL applications in LLMs, reliance on solvers as verifiers, possible overfitting or data bias, benchmark cleanliness, and incomplete open-sourcing at submission time. Post-rebuttal comments, however, acknowledged that the authors’ clarifications strengthened the paper, with some reviewers moving to acceptance while others stayed borderline due to empirical rigor and novelty concerns.

Overall, the paper presents a strong and well-validated contribution, and despite some concerns about benchmark quality and incremental novelty, the clear improvements, thoughtful design, and solid rebuttal justify acceptance.